# Linear ubiquitination by LUBEL has a role in *Drosophila* heat stress response

Tomoko Asaoka[1], Jorge Almagro[1], Christine Ehrhardt[1], Isabella Tsai[2], Alexander Schleiffer[1,3], Luiza Deszcz[1], Sini Junttila[4], Leonie Ringrose[1,5], Karl Mechtler[1,3], Anoop Kavirayani[4], Attila Gyenesei[4], Kay Hofmann[6], Peter Duchek[1], Katrin Rittinger[2] & Fumiyo Ikeda[1,*]

## Abstract

The HOIP ubiquitin E3 ligase generates linear ubiquitin chains by forming a complex with HOIL-1L and SHARPIN in mammals. Here, we provide the first evidence of linear ubiquitination induced by a HOIP orthologue in *Drosophila*. We identify *Drosophila* CG11321, which we named Linear Ubiquitin E3 ligase (LUBEL), and find that it catalyzes linear ubiquitination *in vitro*. We detect endogenous linear ubiquitin chain-derived peptides by mass spectrometry in *Drosophila* Schneider 2 cells and adult flies. Furthermore, using CRISPR/Cas9 technology, we establish linear ubiquitination-defective flies by mutating residues essential for the catalytic activity of LUBEL. Linear ubiquitination signals accumulate upon heat shock in flies. Interestingly, flies with LUBEL mutations display reduced survival and climbing defects upon heat shock, which is also observed upon specific LUBEL depletion in muscle. Thus, LUBEL is involved in the heat response by controlling linear ubiquitination in flies.

**Keywords** deubiquitinase; linear chain; LUBEL; ubiquitin; ubiquitin E3 ligase
**Subject Category** Post-translational Modifications, Proteolysis & Proteomics

## Introduction

Linear ubiquitin chains play an important role in the regulation of immune responses in mammals [1–5]. They are a unique type of chains in which linkage occurs via methionine 1 (Met 1) instead of classical intrinsic lysine (Lys) residues (Fig EV1A). The only known E3 ligase, which is able to generate linear ubiquitin chains, is a RING in-between RING (RBR)-type E3 ligase called "HOIL-1-inter-acting protein" (HOIP)/RNF31 [6]. In mammals, HOIP forms a ligase complex called the "linear ubiquitin chain assembly complex" (LUBAC) with the regulatory subunits "Heme-oxidized IRP2 ubiqui-tin ligase-1" (HOIL-1L)/Rbck1 and SHARPIN/SIPL1, which are both required for full ligase activity of HOIP [7–9]. The RBR domain of HOIP constitutes the catalytic center of LUBAC, and the linear ubiquitin chain determining domain (LDD) at the C-terminus of HOIP is critical for specifying the Met 1 linkage type [10,11].

Recent studies clarified that HOIP is important in regulating the tumor necrosis factor (TNF)-dependent nuclear factor-kappaB (NF-κB) signaling pathway [12,13], the TNF receptor (TNFR) complex II-mediated anti-apoptosis pathway [8,14,15], and the NOD2 innate immune signaling pathway [16]. HOIP was found to play an essential role during mouse embryonic development and cell death, based on phenotypes of HOIP-deficient mice and catalyti-cally dead HOIP knockin mice [14,17]. Taken together, these find-ings suggest that HOIP-dependent linear ubiquitination plays a critical role in development and immune response signaling path-ways. However, its physiological roles *in vivo* remain largely elusive.

Deubiquitinases (DUBs) play a critical role in the negative regula-tion of ubiquitination. In the case of linear ubiquitination, two DUBs called ovarian tumor (OTU) DUB with linear linkage specificity (OTULIN) and cylindromatosis (CYLD) interact with the PUB domain at the N-terminus of HOIP, and differentially regulate LUBAC-dependent signaling pathways [18–22]. Biochemically, OTULIN hydrolyzes linear ubiquitin chains specifically. Ectopic expression of OTULIN in HEK293 and HeLa cells inhibits LUBAC-dependent NF-κB signaling, suggesting that OTULIN is a negative regulator of this signaling pathway [18,20]. In contrast, CYLD hydrolyzes Lys 63-linked and linear ubiquitin chains. Similar to OTULIN, CYLD is also a negative regulator of the NF-κB signaling pathway [23].

To understand the functional roles of linear ubiquitination in biology, we first aimed to establish a linear ubiquitination-defective animal model. As previously mentioned, HOIP knockout and catalytically dead HOIP knockin mice are embryonic lethal. Further-more, in mammals, there are two regulatory subunits of LUBAC,

1  Institute of Molecular Biotechnology of the Austrian Academy of Sciences (IMBA), Vienna, Austria
2  Molecular Structure of Cell Signalling Laboratory, The Francis Crick Institute, London, UK
3  Research Institute of Molecular Pathology (IMP), Vienna, Austria
4  Vienna Biocenter Core Facilities GmbH (VBCF), Vienna, Austria
5  Humboldt-Universität zu Berlin IRI for the Life Sciences, Berlin, Germany
6  Institute for Genetics, University of Cologne, Cologne, Germany
   *Corresponding author. Tel: +43 1 79044 4900; E-mail: fumiyo.ikeda@imba.oeaw.ac.at

SHARPIN and HOIL-1L, which play an essential role in linear ubiquitination and maintaining protein stability of the LUBAC components [7–9]. To determine the specific function of HOIP and HOIP-dependent linear ubiquitination *in vivo*, we searched for species where only the catalytic E3 ligase HOIP is conserved but not the SHARPIN or HOIL-1L subunits. We uncovered *Drosophila CG11321* as an orthologue of mammalian HOIP and found that *Drosophila* is the only species where HOIP is conserved but not HOIL-1L or SHARPIN. Here, we provide the first evidence of linear ubiquitination in *Drosophila* dependent on CG11321, which we named "Linear Ubiquitin E3 ligase" (LUBEL). We show that loss of LUBEL activity renders flies sensitive to heat shock, suggesting a novel function of linear ubiquitination by LUBEL in heat tolerance.

## Results

### *Drosophila* CG11321 is an orthologue of mammalian HOIP

We first aimed to identify ubiquitin E3 ligases responsible for linear ubiquitination in different species. We found that HOIP is highly conserved from *H. sapiens* to *X. laevis* at the amino acid sequence level (Fig EV1B–F). The LUBAC cofactors, HOIL-1L and SHARPIN, are also conserved in these species, but not in *Drosophila melanogaster* (*D. melanogaster*). We did identify CG11321 (Linear Ubiquitin E3 ligase, LUBEL) as a larger orthologue of HOIP in *D. melanogaster* (Fig 1A), which contains many conserved domains. In particular, the catalytic C-terminal region containing the RBR and LDD domains (RBR-C) is highly conserved in these different species (Fig EV1B). For example, critical residues such as the zinc ion-coordinating residues or the Cys residue that forms a thioester intermediate with ubiquitin are perfectly conserved (Fig EV1B). However, LUBEL does not contain the N-terminal PUB domain found in HOIP, which forms a critical interaction with the deubiquitinase (DUB) OTULIN [19–21]. Furthermore, *Drosophila* does not contain an orthologue of human OTULIN, suggesting that other DUBs regulate deubiquitination of linear ubiquitin chains in flies.

Interestingly, LUBEL contains two ubiquitin-associated domains (UBAs) compared to mammalian HOIP, which contains only one. LUBEL contains a HOIP family-specific UBA (UBA1)-like mammalian HOIP, as well as a classical UBA (UBA2) (Figs 1A and EV1C and D) [24]. We performed GST-pulldown assays to determine whether the LUBEL-UBAs interact with Lys 48-linked, Lys 63-linked, or linear ubiquitin chains (Fig 1B and Appendix Fig S1). We found that LUBEL-UBA1 did not interact with any of the ubiquitin chains linked via Lys 48 or Lys 63, or linear ubiquitin chains (Appendix Fig S1), as previously observed with the UBA from human HOIP [8]. On the other hand, LUBEL-UBA2 interacted with tetra or longer poly-ubiquitin chains (Appendix Fig S1), di-ubiquitin chains linked via Lys 63 and to a lesser extent, with chains linked via Lys 48 (Fig 1B). Thus, LUBEL-UBA2 can interact with ubiquitin chains, in particular with Lys63-linked chains, and is distinct from the UBA in mammalian HOIP and from LUBEL-UBA1.

The N-terminus of LUBEL contains a B-box domain and an Npl4 zinc finger (NZF) domain (Figs 1A and EV1E and F). One of the two

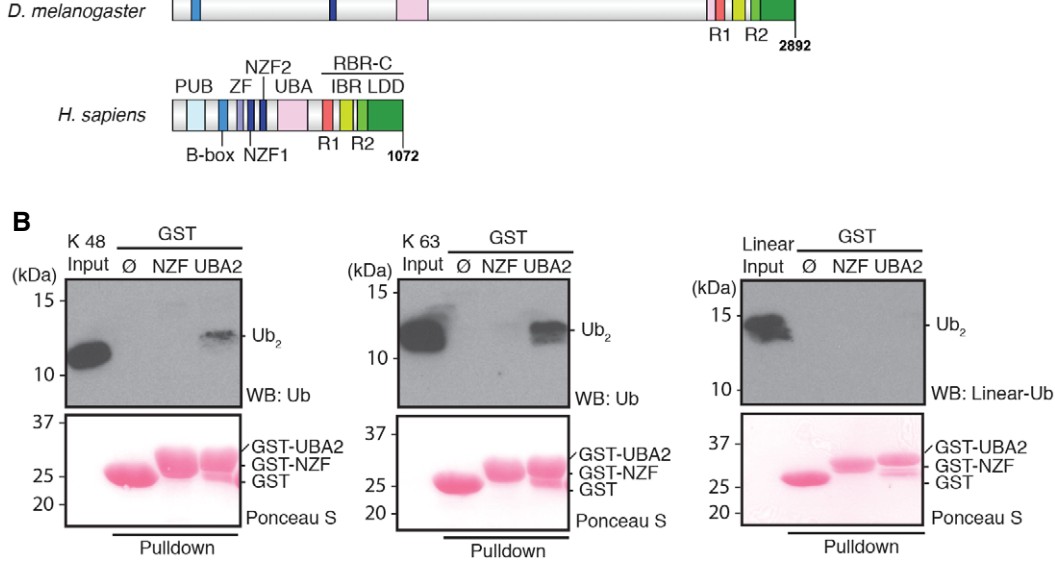

**Figure 1.   Conservation of functional domains of HOIP in *Drosophila* CG11321.**

A   Schematic diagram of HOIP orthologues in *Drosophila melanogaster* (*D. melanogaster*) and *Homo sapiens* (*H. sapiens*). *Drosophila melanogaster* CG11321 is predicted to contain B-box (blue), Npl4 zinc finger (NZF) (navy), ubiquitin (Ub)-associated (UBA1 and UBA2) (pink), and a C-terminal catalytic region, RING between RING (RBR)-C. The RBR-C region consists of RING1 (R1; red), in-between RING (IBR; yellow), RING2 (R2; light green), and linear Ub chain determining domain (LDD; green). PNGase/Ub-associated (PUB) (sky blue) and zinc finger (ZF) (lavender) domains were found in *H. sapiens* but not in *D. melanogaster*.

B   Ub chain interaction of the LUBEL-NZF and UBA2 domains. Immobilized GST-NZF or GST-UBA2 was incubated with Lys (K) 48-, K 63-linked, or linear di-Ub (Ub$_2$) chains and bound Ub chains were subjected to immunoblotting using anti-Ub antibody or a linear linkage-specific Ub antibody (anti-linear Ub antibody). Loading of GST-fusion proteins was visualized by Ponceau S staining.

NZFs in human HOIP, NZF1, is a ubiquitin-interacting domain [8]; thus, we tested whether the LUBEL-NZF domain interacts with ubiquitin chains by performing a GST-pulldown assay. In contrast to LUBEL-UBA2, we did not detect any interaction between LUBEL-NZF and Lys 48-, Lys 63-linked, or linear di-ubiquitin chains (Fig 1B), besides a very weak interaction with linear tetra-ubiquitin chains (Appendix Fig S1).

In summary, LUBEL interacts with ubiquitin chains via UBA2 whereas mammalian HOIP uses NZF1 for ubiquitin chain binding.

## LUBEL catalyzes linear ubiquitin chain formation *in vitro*

To determine whether the predicted LUBEL catalytic region can generate linear ubiquitin chains *in vitro*, we purified a recombinant LUBEL catalytic region-containing fragment (LUBEL-RBR-C) from *E. coli* and performed *in vitro* ubiquitination assays. We found that LUBEL-RBR-C generates ubiquitin chains in an ATP-dependent manner in combination with two different *Drosophila* E2s, UbcD10 (closest homologue of human UbcH7) or Effete/UbcD1 (closest homologue of human UbcH5) [25,26] (Fig 2A). Ubiquitin chains did not form when N-terminally His-tagged ubiquitin was used, indicating that an intact Met 1 in ubiquitin is critical for ubiquitin chain formation by LUBEL-RBR-C (Fig EV2A). The linkage types of ubiquitin chains generated by LUBEL-RBR-C were confirmed to be linear/Met 1 by mass spectrometry analysis (Fig EV2B) as well as by immunoblotting using a linear linkage-specific antibody (Figs 2A and EV2A). Furthermore, we determined whether ubiquitin chains generated by LUBEL-RBR-C were recognized by OTULIN, a linear ubiquitin chain-specific DUB, or vOTU, a Lys-linked-specific DUB [18,27,28]. We confirmed that vOTU and OTULIN cleaved only Lys-linked or linear-linked chains, respectively, *in vitro* (Fig EV2C and D). Furthermore, we found that OTULIN, but not a catalytically dead OTULIN mutant (Cys 129 Ala) or vOTU, cleaved the ubiquitin chains generated by LUBEL-RBR-C *in vitro* (Fig 2B). Together, these data indicate that LUBEL-RBR-C generates exclusively linear ubiquitin chains *in vitro*.

Next, we investigated the structure–function requirements of the LUBEL-RBR-C. We substituted the zinc ion-coordinating Cys residues (C2690/C2693) in the RING2 domain of LUBEL-RBR-C (Fig EV1B) with serine (S) to determine whether they are critical for

linear ubiquitination. Unlike wild-type LUBEL-RBR-C, the LUBEL-RBR-C (CC/SS) mutant did not generate ubiquitin chains *in vitro* (Fig EV2E). Similarly, exogenously introduced LUBEL-RBR-C, but not LUBEL-RBR-C (CC/SS), increased the total levels of linear ubiquitin chains in insect Schneider 2 (S2) cells (Fig EV2F). We next determined whether LUBEL-RBR-C catalyzes linear ubiquitin chain formation using a RING/HECT hybrid mechanism, as previously shown for mammalian RBR ligases [29]. Cys 885 in human HOIP forms the thioester intermediate during ubiquitin transfer [10,30]. Thus, we substituted the corresponding, conserved Cys residue in LUBEL, Cys 2704, with Ala. LUBEL-RBR-C (C2704A) displayed drastically reduced linear ubiquitin chain formation compared to wild type (Fig 2A). The residual activity of this mutant may derive from the activities of the RING domains. By monitoring thioester formation with fluorescently labeled ubiquitin, we confirmed that C2704 formed a thioester intermediate and that LUBEL functions as a RING/HECT hybrid E3 ligase similar to human HOIP (Fig 2C) [28]. In mammalian HOIP, the C-terminal LDD domain, especially human HOIP residues R935 and D936, plays an important role in determining the linkage type of ubiquitin chains by presenting the acceptor ubiquitin (Fig EV1B). In accordance, substitutions at the equivalent positions LUBEL, D2755A and especially R2754A, reduced linear ubiquitin chain synthesis (Fig 2D). In contrast to mammalian full-length HOIP, which requires its binding partners HOIL-1L or SHARPIN to generate linear ubiquitin chains, transient expression of full-length LUBEL alone in S2 cells was sufficient for ubiquitin chain formation (Fig 2E). This observation was supported by the *in vitro* ubiquitination assay using recombinant full-length LUBEL purified by a baculovirus expression system (Fig EV2G). Because the UBA2 domain in LUBEL is localized adjacent to the RING1 domain, we examined whether UBA2 may affect the catalytic activity of LUBEL-RBR-C. We found that adding UBA2 to the N-terminus of RBR-C did not alter its activity *in vitro* (Fig EV2H), suggesting that UBA2 does not contribute to the catalytic activity of LUBEL, at least *in vitro*.

## Adult flies and *Drosophila* S2 cells contain endogenous linear ubiquitin chains

Once we established LUBEL to be a functional E3 ligase for linear ubiquitination *in vitro*, we next aimed to analyze whether

**Figure 2. LUBEL catalyzes the synthesis of linear Ub chains.**

A   *In vitro* ubiquitination assays of LUBEL-RBR-C WT or catalytically dead C2704A in combination with Ube1 and two different *Drosophila* E2s, UbcD10, or Effete/UbcD1. Reactions were terminated at indicated times, and synthesized Ub chains were detected by immunoblotting using anti-linear Ub antibody. Total protein loading was visualized by Ponceau S staining. *: nonspecific band.

B   *In vitro* deubiquitination of Ub chains synthesized by HOIP- and LUBEL-RBR-C. Ub chains produced by RBR-C (human or *Drosophila*) with UbcH7, UbcD10 or Effete were incubated with a linear linkage-specific DUB, OTULIN (WT or catalytically dead C129A (C/A) mutant), or a Lys-linkage-specific DUB, vOTU. The DUB-treated samples were subjected to Coomassie staining or immunoblotting using anti-Ub antibody. *: nonspecific band.

C   Thioester formation assay using Atto647-labeled Ub. Ubiquitination assay using LUBEL-RBR-C WT (left panel) or C2704A (right panel): lanes 1/2 Atto-Ub + E1, lanes 3/4 + ATP, lanes 5/6 + E2, lanes 7/8 + E3, and lanes 9/10 + Ub-His$_6$. In the presence of N-terminally tagged Atto-Ub and C-terminally tagged Ub-His$_6$, only Ub$_2$ product can be synthesized and longer chain formation is restricted. Samples are run without or with DTT in odd or even numbered lanes, respectively. The gels monitor the Atto-labeled Ub.

D   The linear Ub chain formation activity of LUBEL-LDD mutants. Recombinant proteins of RBR-C WT, C2704A mutant, and LDD mutants (R2754A and D2755A) were assessed for their activity by *in vitro* ubiquitination assay. The samples were resolved on a gel and stained with Coomassie or immunoblotted using anti-linear Ub antibody.

E   Linear Ub chain formation by full-length LUBEL transient expression in insect cells. Full-length Myc-LUBEL was transiently expressed in *Drosophila* Schneider 2 (S2) cells, and Myc-HOIP alone or with HA-HOIL-1L in HEK293T cells. Total cell lysates (TCL) of control and transfected samples were incubated with immobilized GST-Linear-Tandem Ub binding entity (Linear-TUBE) containing three tandem repeats of ABIN-1-UBAN. Pulldown samples were blotted with anti-linear Ub antibody, while TCL were blotted with anti-Myc antibody for exogenous LUBEL and HOIP, anti-HA antibody for HOIL-1L, and anti-tubulin antibody for loading. Input of GST proteins was analyzed by Ponceau S staining. *: nonspecific band.

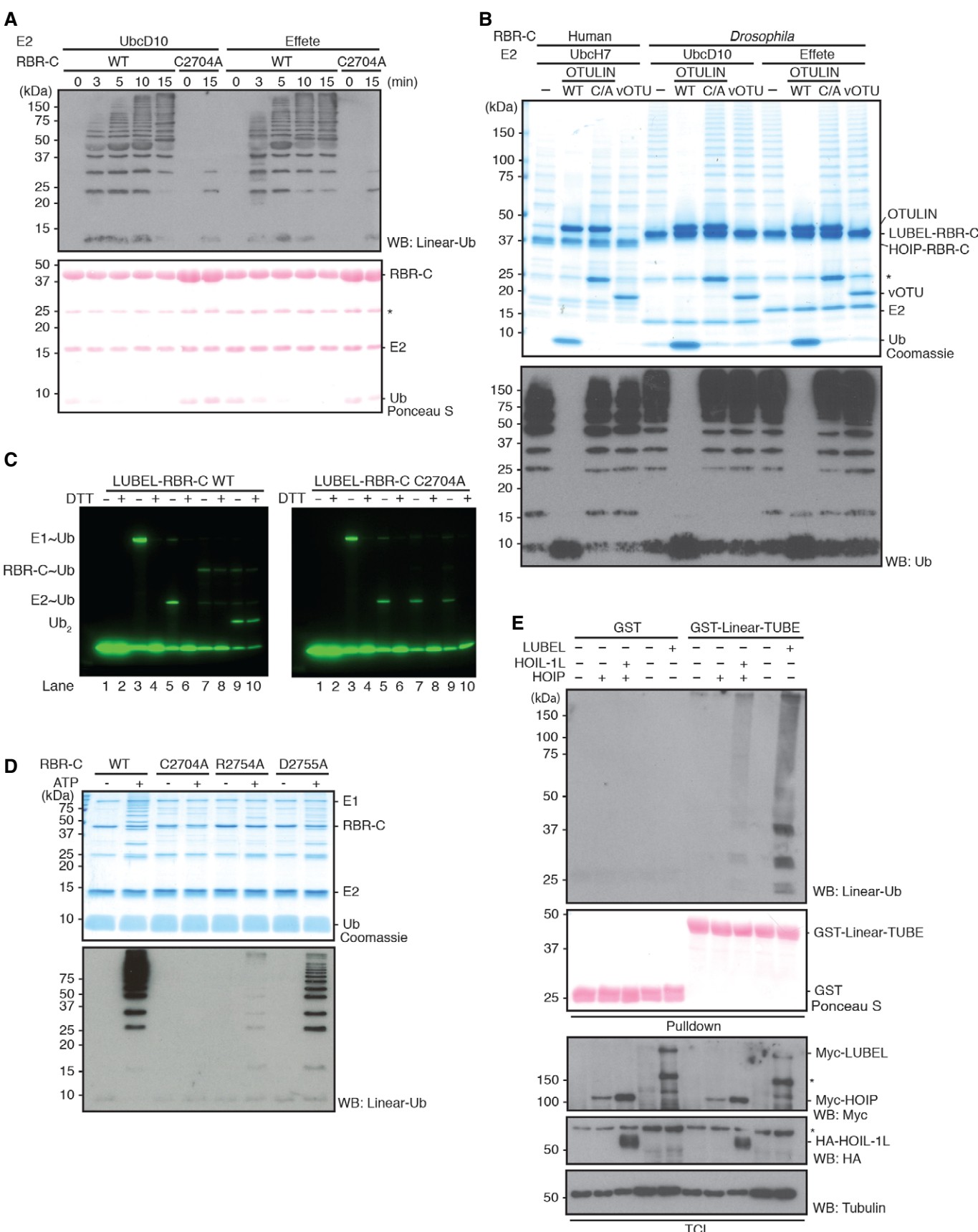

**Figure 2.**

endogenous linear ubiquitination occurs in *Drosophila*. For this purpose, we utilized linear ubiquitin chain-specific tandem-repeated ubiquitin-binding entities (Linear-TUBE), which recognize linear ubiquitin chains through tandem repeats of three ABIN-1-UBAN domains [8,31–33]. We detected linear ubiquitin chains in nondenatured adult fly lysates by combining Linear-TUBE-dependent enrichment with a linear ubiquitin-specific antibody for immunoblotting (Fig 3A). To confirm that the ubiquitin chains we detected are linked linearly, we incubated Linear-TUBE-enriched nondenatured fly lysates with vOTU or OTULIN DUBs, which, respectively, cleave

Lys-linkages and linear-linkages, as described above. As expected, OTULIN treatment abolished the detection of ubiquitin chains (Fig 3A, top panel), confirming that linear ubiquitination occurs in adult flies. Interestingly, vOTU treatment partially reduced the linear ubiquitination signal at high molecular weight (Fig 3A, top panel), and significantly suppressed the total ubiquitin signal (Fig 3A, middle panel), suggesting that the Linear-TUBE-enriched fractions contain mixed linkage types of ubiquitin chains. However, linear ubiquitination signals were not detectable in *Drosophila* S2 cells by Linear-TUBE enrichment due to the low level of linear ubiquitin

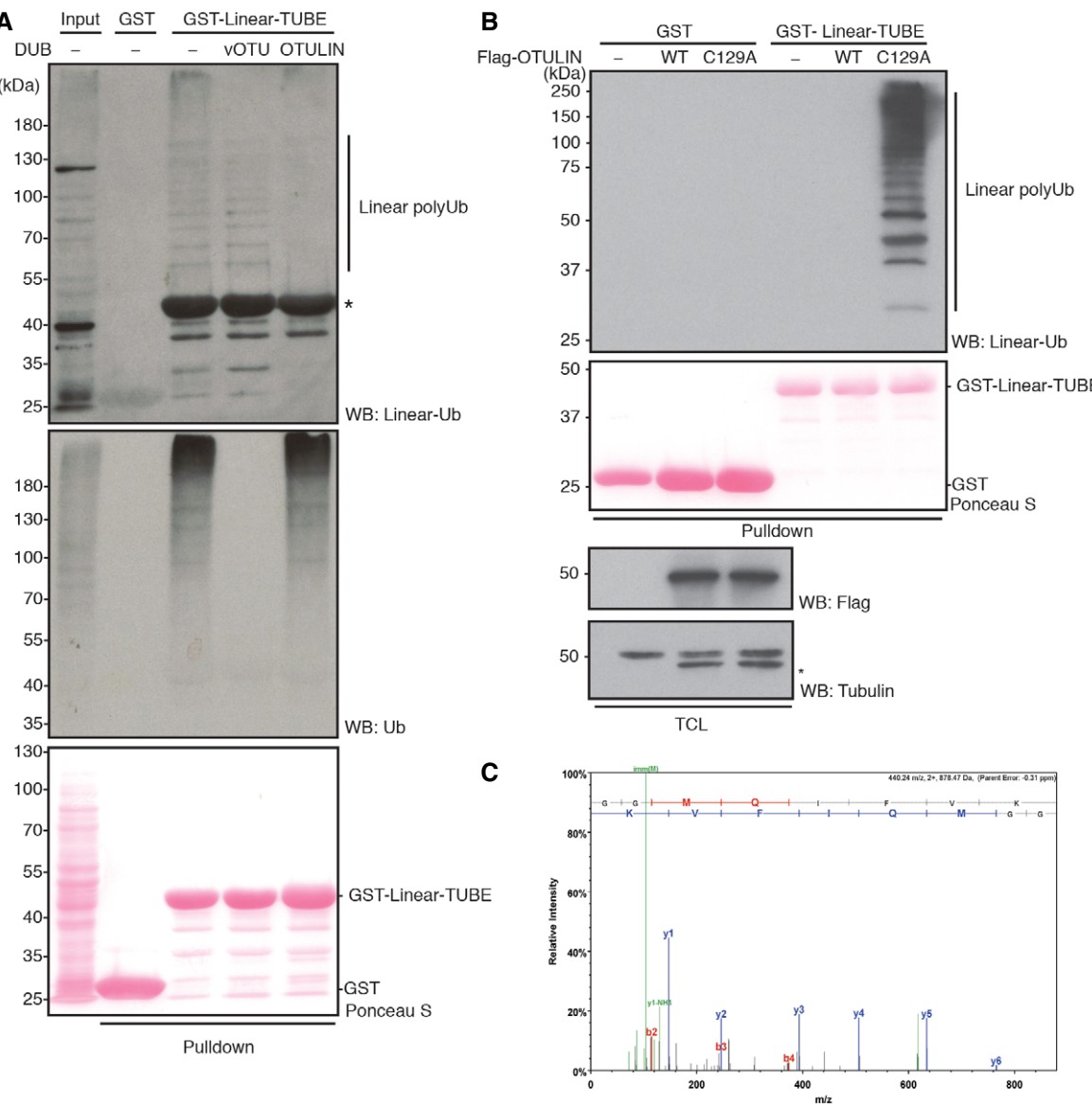

**Figure 3.  Endogenous linear Ub chains are present in *Drosophila*.**

A  Endogenous linear Ub chains enriched in total protein extracts (input, lane 1) from adult w[1118] (w[−]) flies by GST-Linear-TUBE (lane 3). The enriched samples were further treated with recombinant DUBs, vOTU (lane 4), or OTULIN (lane 5). Ub chains were visualized by immunoblotting using anti-linear Ub antibody or anti-Ub antibody. Loading of GST proteins was analyzed by Ponceau S staining. *: nonspecific band.

B  Endogenous linear Ub chains in total cell lysate (TCL) of S2 cells expressing human Flag-OTULIN WT or catalytically inactive C129A mutant. TCLs enriched with GST-Linear-TUBE were examined by immunoblotting using anti-linear Ub antibody. Expression of OTULIN was analyzed by anti-Flag antibody, and loading of TCL was detected by anti-tubulin antibody. Input of GST proteins was analyzed by Ponceau S staining. *: nonspecific band.

C  Mass spectrometry analysis using an identical sample as in panel (B) lane 6. MS/MS spectra of the prototypic linear Ub chain peptide (GGMQIFVK) is shown.

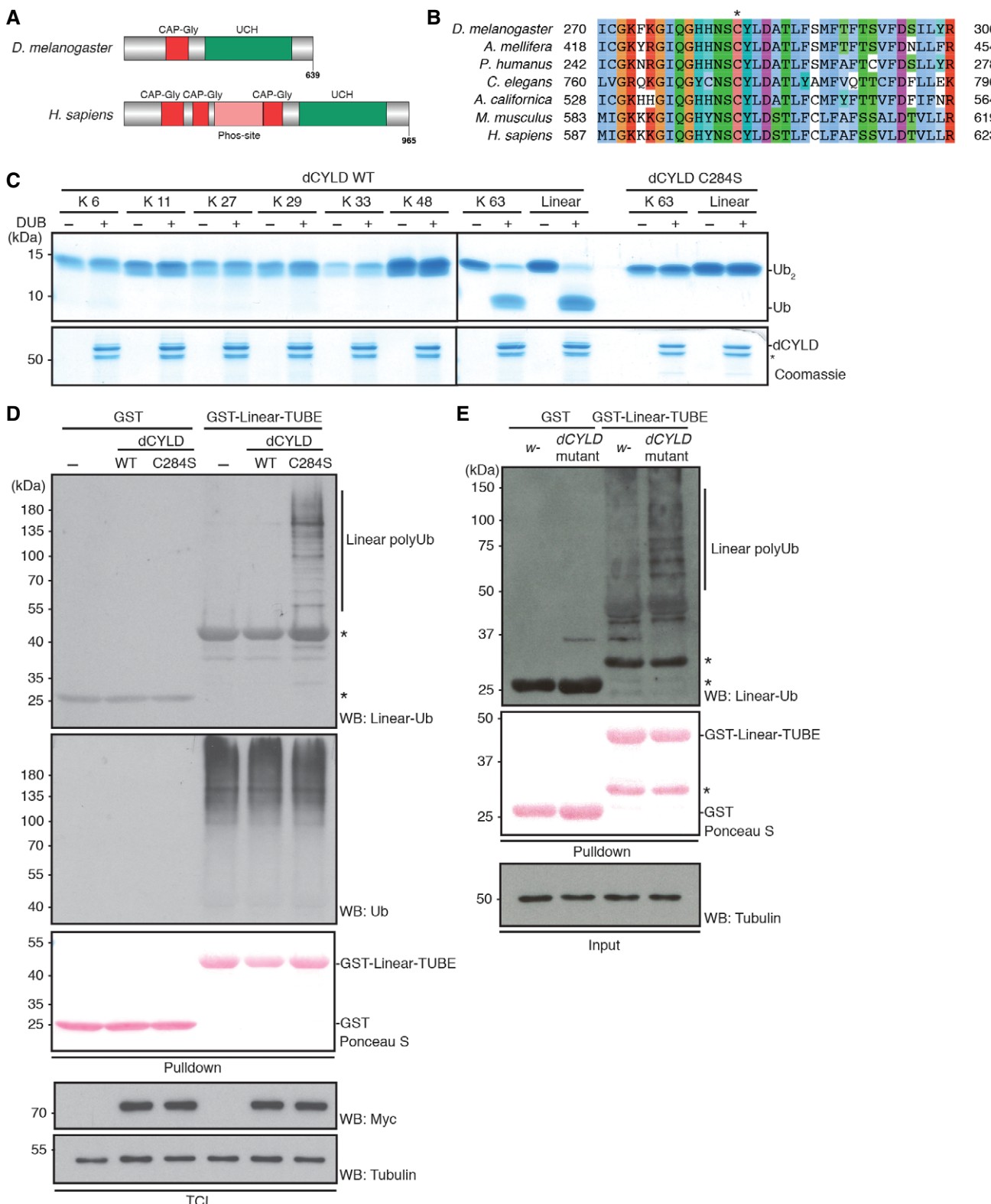

**Figure 4.**

chains (Fig 3B). To overcome this limitation, we took advantage of the catalytically dead OTULIN C129A mutant, which is known to capture linear ubiquitin chains and protect them from cleavage by other DUBs in mammalian cells [18]. Indeed, we detected linear ubiquitin chains in lysates from *Drosophila* S2 cells expressing catalytically dead OTULIN C129A, when enriched with Linear-TUBE matrix (Fig 3B). We analyzed OTULIN C129A-expressing S2 lysates enriched by Linear-TUBE by mass spectrometry and identified unique

**Figure 4.  dCYLD hydrolyzes linear and K 63-linked Ub chains.**

A, B  Schematic diagram of *Drosophila* (*D. melanogaster*) CYLD (dCYLD) and human (*H. sapiens*) CYLD (A), and a multiple amino acid sequence alignment of a catalytic region of the ubiquitin carboxyl-terminal hydrolase (UCH) domain in different species (B). dCYLD is 639 amino acids in length, and contains one cytoskeletal-associated protein–glycine-conserved (CAP-Gly) domain (Red) and the UCH domain (Green) (A). Human CYLD is 965 amino acids in length and contains three CAP-Gly domains, an unstructured phosphorylation region (Phos-site (Pink); specific for CYLD chordata species), and a UCH domain (Green). A conserved active Cys (C284 in *Drosophila*) is indicated by * (B).

C   *In vitro* deubiquitination assay of dCYLD. dCYLD was incubated with all eight linkage types of Ub$_2$ chains. dCYLD catalytically dead C284S mutant was incubated with K 63- or linear Ub$_2$ chains. All the proteins in the reactions were resolved on Coomassie-stained SDS–PAGE gel. *: nonspecific band.

D   Stabilization of linear Ub chains by dCYLD C284S mutant transient expression in S2 cells. Myc-tagged dCYLD WT and C284S were transfected in S2 cells, and linear Ub chains were enriched with GST-Linear-TUBE and immunoblotted by anti-linear Ub or anti-Ub antibodies. Input of GST proteins was visualized by Ponceau S staining. Expression of Myc-dCYLD was examined by anti-Myc antibody, and anti-tubulin antibody was used for the loading control of TCL. *: nonspecific band.

E   Linear Ub chains in *dCYLD* mutant flies. Endogenous levels of linear Ub chains in *dCYLD* mutant flies were compared with a control *w⁻* fly strain by performing GST-Linear-TUBE pulldown. Enriched linear Ub chains were immunoblotted with anti-linear Ub antibody. Input of GST proteins was visualized by Ponceau S staining. Tubulin was used for the loading control of the input. *: nonspecific band.

peptides (GGMQIFVK) derived specifically from linearly linked ubiquitin chains (Fig 3C). To our knowledge, these data provide the first evidence of endogenous linear ubiquitination in *Drosophila*.

### *Drosophila* CYLD specifically hydrolyzes linear and Lys 63-linked ubiquitin chains

Because we did not find orthologues of the linear ubiquitin-specific DUB OTULIN in *Drosophila*, we aimed to identify a DUB that specifically hydrolyzes linear ubiquitin chains. In mammals, it has been shown that CYLD has a dual specificity for linear- and Lys 63-linked ubiquitin chains [34]. *Drosophila* CYLD (dCYLD) is an active DUB [35]; however, the linkage specificity of dCYLD has not been defined. We found that the catalytic core of dCYLD was well conserved among species (Figs 4A and B, and EV3A). Similar to human CYLD, recombinant dCYLD hydrolyzed linear and Lys 63-linked di-ubiquitin chains, but not other types of chains (Lys 6, Lys 11, Lys 27, Lys 29, Lys 33, or Lys 48) (Fig 4C). We further analyzed the dCYLD catalytic region by substituting the catalytic Cys 284 with Ser [35]. As expected, the dCYLD C284S was not able to hydrolyze Lys 63-linked or linear di-ubiquitin chains *in vitro* (Fig 4C). As observed for OTULIN C129A, ectopic expression of dCYLD C284S in S2 cells increased the total linear ubiquitination signal, strongly suggesting that both of these catalytically dead DUBs have dominant negative activity (Fig 4D). Importantly, we observed an increase in the basal level not only of the Lys 63-linked ubiquitination signal, but also of the linear ubiquitination signal in dCYLD-deficient fly lysates compared to control fly lysates (Figs 4E and EV3B). Although LUBEL apparently lacks the PUB domain found in HOIP (Fig 1A), which is important for recruiting a SPATA2-CYLD complex in mammals [36], we could detect an interaction between the LUBEL-RBR-C domain and dCYLD in pulldown experiments (Fig EV3C and D). These data collectively suggest that dCYLD is a negative regulator of linear and Lys 63-linked ubiquitination and may interact with LUBEL in *Drosophila*.

### Linear ubiquitination is abolished in catalytically dead LUBEL mutant fly lines

It was previously shown by *in situ* hybridization that LUBEL mRNA levels increase in muscles during embryogenesis [37–39]. There are seven annotated polypeptides for LUBEL in FlyBase. Because not all the annotated transcripts contain the RBR-C catalytic region, we performed qPCR of samples from different embryonic stages using

primer sets that amplify mRNAs encoding either only the four longer polypeptides containing RBR-C, or three additional polypeptides for the N-terminal region. Using both primer sets, we found that LUBEL mRNA levels increased during embryogenesis, especially at the embryonic stages between 10 and 14 h (Fig EV4A). Furthermore, we detected expression of mRNA corresponding to the longer isoforms, which contain the RBR region, in adult flies and S2 cells by RNA-Seq (Fig EV4B).

To determine the physiological relevance of LUBEL-dependent linear ubiquitin chains *in vivo*, we next aimed to establish catalytically dead LUBEL mutant flies. We used CRISPR/Cas9 technology to establish two different catalytically dead LUBEL mutant lines: in one of the zinc ion-coordinating Cys residues in the RING2 domain are substituted with Ser (*CC/SS*, as analyzed in Fig EV2E and F), and in the other, the RING2 domain is deleted (*delR2*) (Fig 5A). We used the Linear-TUBE matrix to evaluate linear ubiquitination in total fly lysates and observed reduced levels of linear ubiquitin chains in LUBEL catalytic mutant flies compared to control *w⁻* flies (Fig 5B). Thus, LUBEL catalytic activity plays a major role in linear ubiquitination in adult flies. The LUBEL mutant flies are fertile and show no obvious gross phenotype. Interestingly, longevity of LUBEL mutant flies compared to control flies was mildly reduced (Fig 5C); however, muscle morphology and functions were not significantly affected in young adults or in aged flies (Fig 5D, Appendix Fig S2A and B, Movies EV1, EV2 and EV3, and Fig EV4C). These are the first available animal models with a ubiquitous defect in linear ubiquitination, and these flies enable us to examine further the role of LUBEL-dependent linear ubiquitination *in vivo*.

### LUBEL catalytic mutation does not affect immune responses

Mammalian LUBAC plays a critical role in immune responses. To investigate a possible role of linear ubiquitination in fly immune responses, we examined the survival responses of LUBEL and dCYLD mutant flies to Gram-negative (*E. coli*) and Gram-positive (*M. luteus*) bacteria (Fig EV4D–F). In contrast to *E. coli*-infected Relish mutant flies (Rel[E20]), which are known to be susceptible to Gram-negative bacterial infection [40], LUBEL mutant flies (*CC/SS* and *delR2*) displayed similar survival rates as parental flies upon bacterial infection by *E. coli* (Fig EV4D). Similarly, loss of LUBEL activity did not affect the survival rate upon *M. luteus* infection (Fig EV4F). The survival rates of dCYLD mutant flies were also similar to LUBEL mutant and parental flies; a minor decrease was observed by *E. coli* infection but not by *M. luteus* infection (Fig EV4E and F).

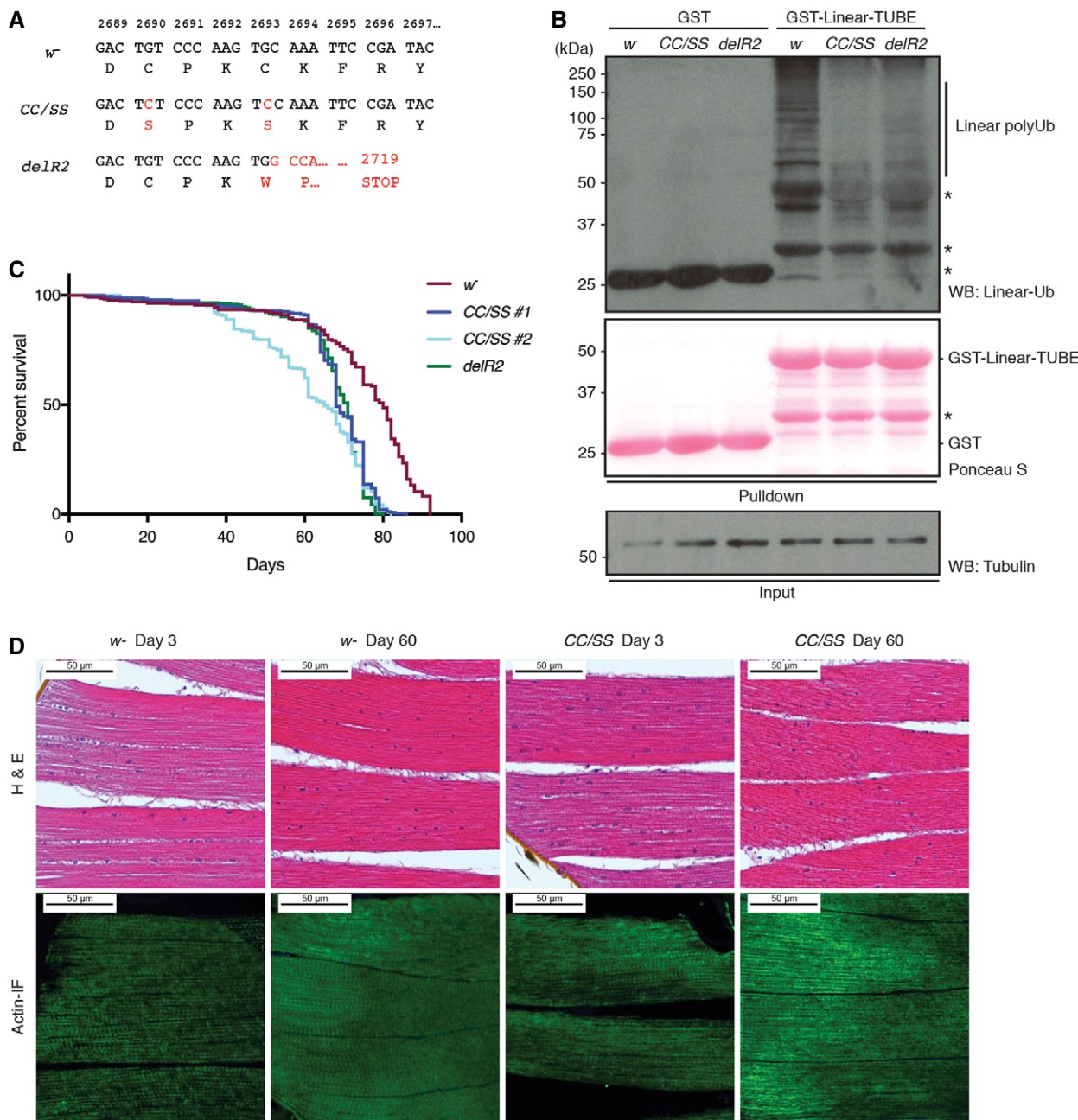

**Figure 5.  Endogenous linear Ub chains are diminished in catalytically dead LUBEL mutant flies.**

A   Amino acid sequences of the mutations introduced in *LUBEL* by CRISPR/Cas9. Two LUBEL mutant lines were created for *in vivo* studies; *w⁻* parental line, *C2690S/C2693S* (*CC/SS*), and *delR2*, which has an indel mutation with an early stop codon at amino acid 2,719 due to a frameshift. Mutations are indicated in red.

B   Linear Ub chains in *w⁻* and LUBEL mutant fly lysates. *w⁻*, *CC/SS*, and *delR2* flies were analyzed for the level of endogenous linear Ub chains by enriching the linear ubiquitin chains by GST-Linear-TUBE. Pulldown samples were immunoblotted by anti-linear Ub antibody, while GST proteins were analyzed by Ponceau S staining. Anti-tubulin antibody was used for the loading control of the input. *: nonspecific band.

C   The life spans of female *w⁻* and catalytically dead LUBEL mutant flies (two clones of *CC/SS*, indicated as *#1* and *#2*, and *delR2*). Survival of three independent cohorts with ~80 flies each was monitored over time. Median survival time (days): *w⁻* = 78, *CC/SS #1* = 68, *CC/SS #2* = 65, and *delR2* = 71. Total sample sizes are as follows: *w⁻* = 215, *CC/SS #1* = 242, *CC/SS #2* = 233, *delR2* = 273. Log-rank (Mantel–Cox) test between fly lines: *w⁻* and *CC/SS #1* > 0.0001, *w⁻* and *CC/SS #2* > 0.0001, *w⁻* and *delR2* > 0.0001, *CC/SS #1* and *CC/SS #2* = 0.0007, *CC/SS #1* and *delR2* = not significant, *CC/SS #2* and *delR2* = not significant.

D   Histological analysis of muscle in LUBEL mutant flies. Hematoxylin and eosin (H&E) (top panels) and actin immunofluorescent staining (actin-IF) (bottom panels) of thorax muscles from young (day 3) or aged (day 60) *w⁻* or *CC/SS* female flies. Scale bars: 50 μm.

We further determined whether Gram-negative bacterial infection affected the expression of anti-microbial peptide genes in LUBEL mutant lines by RNA-Seq at the 2-h time point (Appendix Tables S1 and S2) and qPCR at the 10-h time point (Fig EV4G). A *Minos*-element mutant (Mi-el) that contains a transposon insertion between the LUBEL-UBA1 and UBA2 was included

in the RNA analysis as an additional LUBEL mutant line. We did not observe significant differences in global gene expression between the parental and mutant flies. Statistical analysis of the RNA-Seq data revealed that only 11 and 36 immune responsive genes of 402 were differentially expressed between any of the control and *E. coli*-pricked sample groups, respectively (Appendix Tables S1 and S2). Although some immune responsive genes were expressed at different levels upon bacterial infection in different fly lines, functional enrichment analyses of all of the genes differentially expressed in the mutant fly lines compared to the control fly line revealed that "immune response" Gene Ontology term was statistically and significantly enriched in only one of the comparisons (Appendix Tables S3–S8). Indeed, the majority of the immune responsive genes were expressed similarly between the sample groups, indicating that there are no overall significant differences in immune responsive gene induction between control and LUBEL mutant flies. To further investigate the immune responses, we evaluated some well-known target genes by qPCR, such as *Attacin C*, *Diptericin*, and *Drosomycin*. *E. coli*-induced gene expression of these genes were not significantly different between parental and LUBEL mutant flies (*CC/SS* and *delR2*) (Fig EV4G). These results collectively suggest that LUBEL-dependent linear ubiquitination signal does not play an essential role in immune responses to bacteria in adult flies.

**LUBEL-dependent linear ubiquitination deficiency sensitizes flies to heat shock**

Next, we evaluated different stress responses. We found that heat shock of adult flies at 36°C led to the accumulation of linear and Lys 48-linked ubiquitin chains (Fig 6A). As previously shown in yeast and mammalian cells [41], the total amount of ubiquitin chains also increased during heat shock in adult flies (Fig EV5A). We confirmed that heat shock increased the level of one of the ubiquitin coding genes, *Ubi-p63E*, by qPCR (Fig EV5B). Two of the ubiquitin genes, *Ubi-p63E* and *Ubi-p5E*, encode tandem repeats of ubiquitin, and parts of their unprocessed protein products are identical to linear ubiquitin chains. However, we did not observe protein products corresponding to the expected size of the ubiquitin coding genes *Ubi-p63E* (85.8 kDa) and *Ubi-p5E* (60.0 kDa) (Figs 6A and EV5A), suggesting that the observed linear ubiquitin chain signal observed upon heat shock does not arise from these genes.

Importantly, we found that the survival rates of heat-treated LUBEL catalytic mutant flies (*CC/SS* and *delR2*) as well as flies with whole-body knockdown of LUBEL (*UAS-shRNA LUBEL* targeting the catalytic region, crossed with *Tubulin-Gal4*) were significantly decreased compared to the $w^-$ control line or to the heterozygous lines (Fig 6B and C). The knockdown efficiency by *Tubulin-Gal4* in flies was verified by qPCR (Fig EV5C). These data suggest that linear ubiquitin chains have an important role in heat tolerance in flies. We found that dCYLD mutant flies have a similar tolerance to heat shock as control flies (Fig EV5D), suggesting that linear and/or Lys 63-linked ubiquitin chains are not limiting for heat tolerance. To understand further the mechanisms of heat stress responses in flies, we analyzed induction of HSP70, a known heat shock response gene, in LUBEL mutant flies (*CC/SS* and *delR2*). Compared to the control flies, LUBEL mutant flies (*CC/SS* and *delR2*) displayed significantly lower induction of HSP70 mRNA upon heat stress at early time points (*CC/SS* and *delR2*) (Figs 6D and EV5E). These data suggest that suppression of heat-dependent HSP70 induction in LUBEL mutant flies may contribute to heat tolerance defect in these flies.

We observed that heat shock induces climbing defects in catalytically dead LUBEL adult flies (*CC/SS* and *delR2*) (Movie EV4), so we next asked whether these defects derive from muscle-specific deficiency of LUBEL. To this end, we used *Mef2-Gal4* and *24B-Gal4* flies to knockdown LUBEL (Figs 6C and EV5F). Myocyte enhancer factor 2 (Mef2) is a transcription factor highly expressed in fly muscles [42,43]; held out wings (how) is a putative RNA binding protein expressed in mesoderm; and *how*[24B] (*24B*) is its established driver line [44,45]. Interestingly, *UAS-shLUBEL* crossed with *Mef2-Gal4* or *24B-Gal4* flies display significantly decreased survival upon heat shock especially at early time points (Figs 6C and EV5F), suggesting that LUBEL has a critical role in muscle to tolerate heat shock. In addition, muscle-specific LUBEL knockdown flies (*24B-Gal4 > LUBEL*) display defective climbing ability upon heat shock (Movie EV5). However, muscle-specific knockdown of LUBEL led to a milder decrease in fly survival upon heat shock than whole-body LUBEL knockdown, suggesting that LUBEL functions in tissues additional to muscle, especially at a late time point. Collectively, LUBEL-dependent linear ubiquitination plays a role in heat tolerance in flies.

## Discussion

In mammals, linear ubiquitin chains are generated by the E3 ligase complex LUBAC, which plays an important role in the regulation of

**Figure 6.** **Catalytically dead LUBEL flies are susceptible to high temperature.**

A The level of poly-Ub chains in $w^-$ flies after heat shock. $w^-$ male flies were incubated in a 36°C water bath for indicated times and the levels of poly-Ub chains in total protein extracts were compared by immunoblotting using antibodies against linear, K 48-, and K 63-linked Ub chains. Ponceau S staining was used to visualize the protein loading.

B Survival of catalytically dead *LUBEL* mutant flies upon heat shock. Young adult flies (15 males and 15 females) were incubated in a 36°C water bath and immobilized flies were counted over the time indicated. Median survival time (h): $w^-$ = 9.25, *CC/SS* #1 = 5.5, *CC/SS* #2 = 5, *delR2* = 5.75, *CC/SS* #1/+ = 8.5, *CC/SS* #2/+ = 10, *delR2/+* = 8. *P*-values calculated by Gehan–Breslow–Wilcoxon test between fly lines: $w^-$ and *CC/SS* #1, *CC/SS*#2, or *delR2* > 0.0001 (****$P$ < 0.0001), $w^-$ and *CC/SS* #1/+ = 0.0134, $w^-$ and *CC/SS* #2/+ = ns, $w^-$ and *delR2/+* = 0.0002, *CC/SS* #1 and *CC/SS* #1/+ > 0.0001, *CC/SS* #2 and *CC/SS* #2/+ > 0.0001, *delR2* and *delR2/+* > 0.0001. Representative data are shown from six independent experiments.

C Survival of whole-body or muscle-specific LUBEL knockdown flies upon heat shock. shRNA-based knockdown (KD) of LUBEL were driven by *Tub-Gal4* or *Mef2-Gal4* flies. Control fly lines (*Tub-Gal4/+* and *Mef2-Gal4/+*) were used to compare with the KD flies. 15 males and 15 females per each line were used in this assay. Median survival time (h): *Tub-Gal4/+* = 7, *Tub-Gal4 > LUBEL* = 3.5, *Mef2-Gal4/+* = 7.5, *Mef2-Gal4 > LUBEL* = 7. *P*-values calculated by Gehan–Breslow–Wilcoxon test: *Tub-Gal4* < 0.0001 (****), *Mef2-Gal4* = 0.0029 (**). Representative data are shown from three independent experiments.

D Heat-induced mRNA expression of HSP70. $w^-$, *CC/SS* #2, and *delR2* flies were heat treated for 60 min and recovered for 1 h, and mRNA HSP70 was quantified by qPCR. Rp49 was used as a reference, and $w^-$ untreated sample was used as calibrator to calculate the expression ratio. Data are analyzed by two-way ANOVA with multiple comparison, and presented as mean ± SD (*$P$ < 0.01, ***$P$ < 0.001, ****$P$ < 0.0001). Representative data are shown from three independent experiments.

immune responses [1–5]. We identified orthologues of the LUBAC catalytic component HOIP, but not the subunits, HOIL-1L or SHARPIN, in *Drosophila*. Catalytically dead LUBEL mutant flies (*CC/SS* and *delR2*) did not show any obvious developmental defects, in contrast to mice deficient for HOIP or expressing a catalytically dead mutant, which are embryonic lethal [14,17].

Given the requirement of HOIL-1L and SHARPIN for linear ubiquitin formation by LUBAC, it was not clear that LUBEL would be sufficient to support a linear ubiquitination system. We show that LUBEL without potential binding partners is sufficient to generate linear ubiquitin *in vitro* and that catalytically dead LUBEL mutant flies (*CC/SS* and *delR2*) are sensitive to heat shock. Tissue-specific

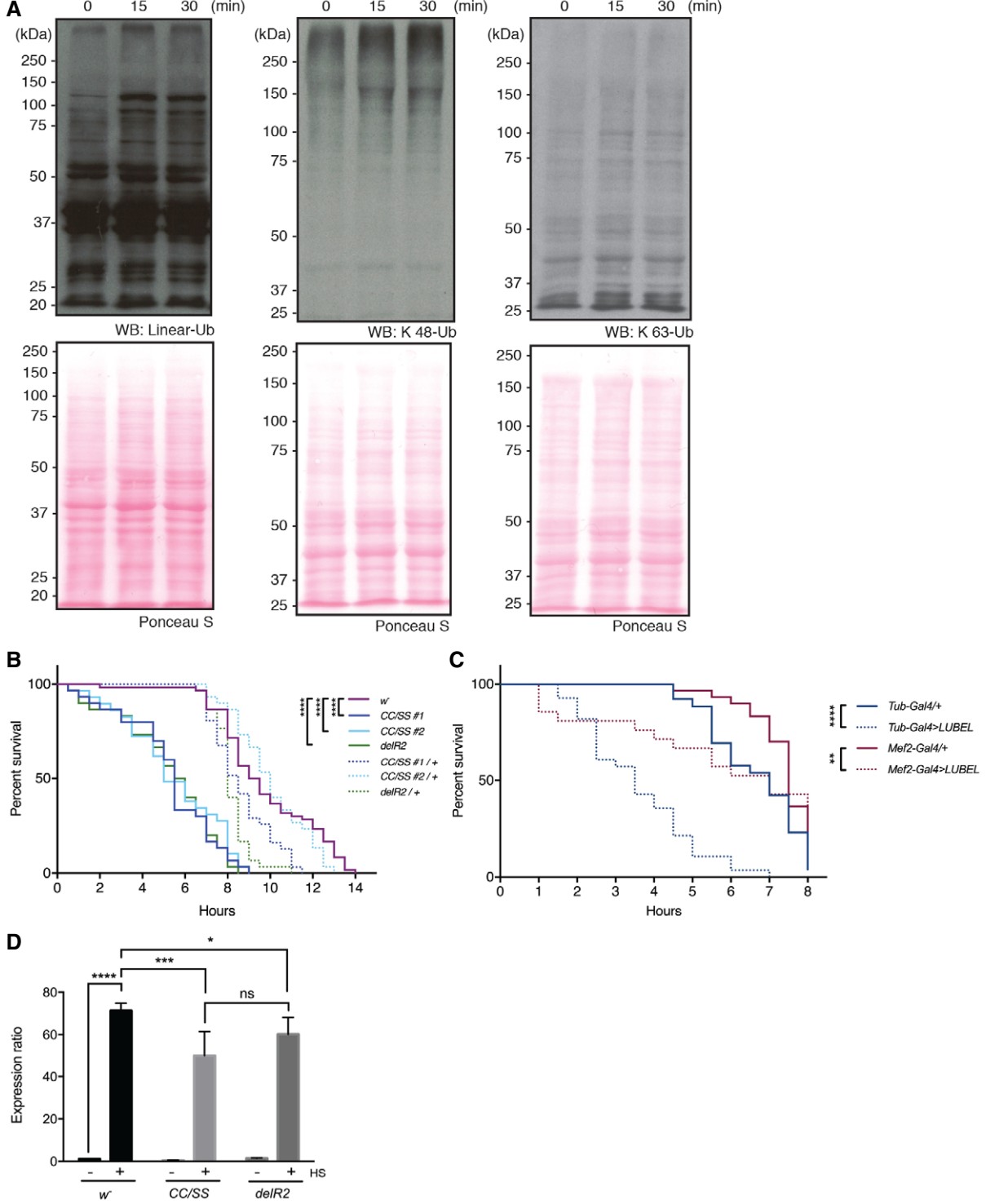

**Figure 6.**

knockdown of LUBEL in muscle also resulted in partial heat tolerance defects. Together with the increased expression of LUBEL during embryogenesis in muscles, our data suggest that LUBEL plays an important role in muscles to regulate heat tolerance. Consistently, we found that heat shock triggered an accumulation of linear ubiquitinated proteins in adult flies. We have not yet identified specific substrates of the LUBEL-dependent linear ubiquitination. However, it is possible that the total amount of linear ubiquitination itself is important for heat tolerance. Based on the observed increased amount of linear ubiquitinated proteins during heat shock, we speculate that heat shock induces LUBEL catalytic activity, or that LUBEL expression increases upon heat shock, generating linear ubiquitin chains in adult flies. In LUBEL mutant flies, heat shock-dependent HSP70 induction was suppressed at early time points after heat shock. Since heat shock response is conserved from yeast to humans [46], linear ubiquitination by HOIP in mammals may also regulate the heat response. HSP70 functions as a molecular chaperone and plays a role in the immune response in mammals [47]; therefore, decrease in HSP70 by LUBAC deficiency may lead to deregulation of immune responses. Interestingly, an Rbck1/HOIL-1L gene mutation was recently identified in myopathy patients [48–51]. Because Rbck1/HOIL-1L deficiency leads to destabilization of other LUBAC components including HOIP and SHARPIN, the myopathy symptoms in these patients may derive from the deregulation of linear ubiquitination. Understanding how LUBEL activity in fly muscle promotes heat tolerance and other functions will be of interest.

At the molecular level, we have shown that linear ubiquitin chain formation in flies depends on the catalytic activity of LUBEL. Similar to the mammalian HOIP catalytic RBR-C domain, the LUBEL-RBR-C domain, together with a *Drosophila* E2 enzyme, specifically generates linear ubiquitin chains *in vitro*. LUBEL contains two UBA domains, and UBA1 is highly similar to HOIP-UBA. We speculate that LUBEL-UBA1 interacts with binding partners containing UBL domains to form a complex, similar to the UBA in human HOIP. Because the known binding partner in mammals, HOIL-1L, does not exist in *Drosophila*, it will be of interest to identify proteins that interact with LUBEL-UBA1. Importantly, in contrast to mammalian HOIP, we observed that full-length LUBEL without potential binding partners is sufficient to generate linear ubiquitin chains in S2 cells and *in vitro*, strongly suggesting that at least for linear ubiquitination, LUBEL-UBA1 does not necessarily require binding partners. Interestingly, LUBEL contains a second UBA domain, UBA2, that is localized adjacent to the RING1 domain and recognizes K 63-linked di-ubiquitin chains, and K 48-linked di-ubiquitin chains to a lesser extent. Despite its proximity, LUBEL-UBA2 does not affect the catalytic activity of LUBEL-RBR-C *in vitro*. It is possible, however, that LUBEL localization in cells may be regulated via UBA2–ubiquitin interactions. Because LUBEL is larger than mammalian HOIP, we speculate that it has unique characteristics. Moreover, there are seven different LUBEL polypeptides annotated and three of them do not contain the UBA2 domain or the catalytic RBR domain. The shorter isoforms of LUBEL may function as adapters for protein–protein interactions in a ubiquitin-independent manner. We demonstrated that whole-body LUBEL knockdown of the longer isoforms in flies phenocopied the sensitivity to heat shock of catalytically dead LUBEL mutant flies. Thus, the catalytic activity of LUBEL is involved in the regulation of heat tolerance.

As with other ubiquitin chain linkage types, linear ubiquitination is tightly regulated by the interplay between the E3 ligase and DUBs. In the case of *Drosophila*, we did not identify any orthologues for mammalian OTULIN, which is thus far the only known DUB specific for linear ubiquitin chains. However, we found that dCYLD specifically hydrolyzes linear and Lys 63-linked ubiquitin chains. These observations suggest that dCYLD functions similarly to mammalian CYLD at the molecular level. In contrast to the known functions of dCYLD in immune responses [35], we did not observe immune response defects in catalytically dead dCYLD mutant flies at least with the bacterial strains we used. Mammalian HOIP does not bind CYLD directly, and instead, the two proteins are linked by SPATA2, which contains a predicted PUB domain in its N-terminus that interacts with CYLD, while its C-terminal portion is recognized by the N-terminus of HOIP [34]. Interestingly, LUBEL lacks a PUB domain, and instead, we could observe a direct interaction between RBR-C and dCYLD in pulldown experiments. These observations suggest that the interaction between LUBEL and dCYLD may occur in a different mode to the one between mammalian HOIP and CYLD. *Drosophila* contains a SPATA2-related gene called *tamo*, which has a SPATA2 homology region at the N-terminus [52]. Tamo was shown to interact with the *Drosophila* Rel transcription factor Dorsal, and to negatively regulate its nuclear import [52]. This is very interesting since the Rel family contains the vertebrate c-Rel oncogene and NF-κB [53]. The possible interplay between LUBEL, dCYLD, Tamo, and Dorsal in the regulation of the fly signaling cascades will need further studies.

To understand whether dCYLD and LUBEL regulate heat tolerance in a cooperative manner, it would be of interest to examine the flies crossed between dCYLD mutant and LUBEL mutant lines. However, linear ubiquitin chains are not formed in LUBEL catalytic mutant flies and the dCYLD mutation would not recover the linear ubiquitination in this condition.

Linear ubiquitin chains are recognized by specific linear ubiquitin-binding proteins. In mammals, there are several proteins, which contain linear ubiquitin chain-specific interaction domains such as NEMO/IKKγ, ABIN proteins, Optineurin, A20, OTULIN, and HOIL-1L [3,26,33,54–56]. Among these proteins, NEMO/IKKγ is conserved in *Drosophila* (called Kenny) and regulates *Drosophila* innate immunity [57–59]. The critical amino acid residues for linear ubiquitin chain recognition by mouse NEMO [54] are conserved in *Drosophila* Kenny. Based on this information, we speculate that Kenny recognizes linear ubiquitin chains and that the linear ubiquitination system is involved in the regulation of *Drosophila* immunity. However, LUBEL catalytic mutant flies and parental flies displayed comparable survival upon infection by Gram-negative and Gram-positive bacteria. Hence, LUBEL-dependent linear ubiquitination does not appear to play an essential role in immunity of adult flies.

In conclusion, we identified a linear ubiquitination system in *Drosophila* and elucidated its novel role in heat tolerance. The establishment of catalytically dead LUBEL mutant flies, which lack linear ubiquitination, enables us to analyze further the functions of linear ubiquitination in the whole body and/or specific tissues of flies. Understanding the novel functions of linear ubiquitination in a specific tissue can be further investigated with tissue-specific knockout mouse models.

# Materials and Methods

### Antibodies and reagents

Antibodies used in this study are as follows: anti-Myc antibody (clone 9E10; Covance, Princeton, NJ), anti-Flag antibody (clone M2; Sigma, St. Louis, MO), anti-HA antibody (HA.11 clone 16B12, Covance, Princeton, NJ), anti-Alpha-tubulin antibody (ab15246, Abcam, Cambridge, UK), anti-Linear Ub antibody (LUB9; Life Sensors, Malvern, PA), anti-Lys 63-Ub antibody (Apu3; Merck Millipore, Darmstadt, Germany), anti-Lys 48-Ub antibody (Apu2; Merck Millipore), and anti-Ub antibody (clone P4D1; Santa Cruz Biotechnology, Santa Cruz, CA). A rabbit polyclonal antibody recognizing LUBEL-RBR was generated using recombinant LUBEL-RBR (aa 2,514–aa 2,655) (immunoGlobe, Himmelstadt, Germany). $His_6$ Ube1, ubiquitin, $His_6$-ubiquitin, di-ubiquitin chains (Lys 6, Lys 11, Lys 27, Lys 29, Lys 33, Lys 48, Lys 63-linked and linear chains), poly-ubiquitin chains (2–7, Lys 48 and Lys 63-linked chains, and tetra linear chains) were purchased from Boston Biochem (Cambridge, MA).

### Plasmids

Following inserts were amplified by PCR from *Drosophila melanogaster* w[1118] male cDNA and cloned in pGex6P1 vector using a standard subcloning method: LUBEL domain regions of NZF (aa 733–aa 762), UBA1 (aa 1,042–aa 1,187), UBA2 (aa 2,457–aa 2,513), UBA2-RBR-C (aa 2,457–aa 2,892), and RBR-C (aa 2,512–aa 2,892), dCYLD, UbcD10, and Effete/UbcD1. TR-TUBE (a gift from Keiji Tanaka, Tokyo Metropolitan Institute of Medical Science, Japan) [60] and Linear-TUBE using mouse ABIN1-UBAN (aa465–aa525) [33] were subcloned in pGex6P1 and pGex4T1, respectively. For insect cell expression, LUBEL-RBR-C (WT or C2690S/C2693S), dCYLD (WT or C284S), and OTULIN (WT or C129A) were subcloned into pAMW and pAFW (gifts from Stefan Ameres, IMBA, Vienna, Austria) by standard subcloning method or by gateway cloning using pENTR/D-TOPO and Gateway LR Clonase II (Invitrogen, Carlsbad, CA). To clone *CG11321* isoE in pAMW, DNA fragments were amplified by PCR and Gibson assembly was used to synthesize full-length LUBEL (see Appendix Table S9 for primers used). For baculovirus-based insect expression, full-length LUBEL was subcloned into a modified version of the pACEBac1 vector (Geneva Biotech). Mutants including LUBEL-RBR-C (C2690S/C2693S), LUBEL-RBR-C (C2704S), LUBEL-RBR-C (R2745A), LUBEL-RBR-C (D2755A), dCYLD (C284S), and OTULIN (C129A) were generated by site-directed mutagenesis. pOPINK-vOTU (CCHFV OTU, aa1–183) was a gift from David Komander (Addgene plasmid # 61589) [28]. Human OTULIN cDNA-containing plasmid was purchased from OriGene (Rockville, MD). pGex6P1-UbcH7 and pET49b-human HOIP-RBR-C were used elsewhere [30]. pcDNA3-Myc-human HOIP and pcDNA3-HA-human HOIL-1L were described previously [8,13].

### Cell culture and transfections

*Drosophila* S2 cells (a gift from Stefan Ameres, IMBA, Vienna) were cultured at 25°C in Schneider's *Drosophila* medium (Sigma), supplemented with 10% fetal calf serum (Thermo Fisher Scientific, Waltham, MA). Human embryonic kidney (HEK) 293T cells (ATCC)

were maintained at 37°C, in 5% $CO_2$ condition in Dulbecco's modified Eagle's medium (Sigma) supplemented with 10% fetal calf serum (Thermo Fisher Scientific), L-glutamine (Sigma), and 100 U/ml penicillin–streptomycin (Sigma). Transfections were performed using X-tremeGENE 9 DNA Transfection Reagent (Roche, Indianapolis, IN) for S2 cells and GeneJuice (Merck Millipore) for HEK293T cells, according to the manufacturer's recommendations.

### Immunoblotting

Adult fly lysates were prepared by crushing 5–10 flies in SDS sample buffer containing β-mercaptoethanol (β-ME) [8] and boiling at 95°C for 5 min. For immunoblotting, the method is described elsewhere [8]. Briefly, recombinant proteins or total lysates were resolved by SDS–PAGE, and proteins were transferred to nitrocellulose membrane (GE Healthcare, Little Chalfont, UK). Membranes were blocked with 5% BSA-TBS and blotted with indicated antibodies in 5% BSA-TBS at 4°C overnight. Goat anti-Mouse IgG-HRP (Bio-Rad, Hercules, CA) or anti-Rabbit IgG-HRP (Dako, Glostrup, Denmark) secondary antibodies were used to visualize proteins by Western Blotting Luminol Reagent (Santa Cruz) on high-performance chemiluminescence films (GE Healthcare, Little Chalfont, UK). Where appropriate, Ponceau S staining was used to visualize transferred proteins on the membranes.

### Protein purification

A method for GST-protein purification is described elsewhere [8]. Briefly, proteins were expressed in *E. coli* BL21 (DE3) (Agilent, Santa Clara, CA) or Rosetta (DE3) (Invitrogen) overnight at 18°C and purified by affinity chromatography using GSTrap HP column (GE Healthcare). For the cleavage of the GST tag on LUBEL proteins, dCYLD, OTULIN, vOTU, and E2 enzymes, PreScission Protease (GE Healthcare Life Sciences) was used. A recombinant full-length LUBEL protein was generated using a baculovirus-based method via incorporation of the transfer plasmids into the EmBacY bacmid (Geneva Biotech) and subsequent transfection into Sf9 cells. The V0 virus was amplified once to generate a V1 virus stock, which was used to infect larger cultures of Sf9. Expression was performed at 27°C by infecting cells at a density of $2 \times 10^6$ cells/ml and harvesting cells 48 h after growth arrest. For the protein purification, Sf9 cell pellets were resuspended in 50 mM HEPES buffer, pH 7.5, 150 mM NaCl, Benzonase, and Complete Protease inhibitors (Roche). Cells were centrifuged for 30 min at 4°C. The soluble anti-washed (50 mM HEPES, pH 7.5, 150 mM NaCl). LUBEL was eluted (50 mM HEPES, pH 7.5, 150 mM NaCl, 2.5 mM desthiobiotin), and concentrated using a Vivaspin concentrator with a MWCO of 100 kDa. ProTech service was performed by the VBCF Protein Technologies Facility (Vienna, Austria).

### GST-TUBE pulldown assay

GST-fused proteins were immobilized on glutathione–Sepharose beads (GE Healthcare) as previously described [8]. S2 cells or adult flies were lysed with chilled lysis buffer (50 mM Tris–HCl, pH 7.5, 100 mM NaCl, 5 mM $MgCl_2$, 10% glycerol, 0.2% NP-40, 1 mM phenylmethanesulfonyl fluoride (PMSF), 20 mM N-ethylmaleimide (NEM) (all from Sigma), and Complete protease inhibitors), and

cleared by centrifugation for 20 min at 4°C. HEK293T cells were lysed with chilled lysis buffer (50 mM HEPES, 150 mM NaCl, 1 mM EDTA, 1 mM EGTA, 25 mM NaF, 10 mM $ZnCl_2$, 10% glycerol, 1% Triton X-100, 20 mM NEM, and complete protease inhibitors). Cleared lysates were subjected to GST pulldown at 4°C overnight using GST-control, GST-Linear-TUBE, or GST-TR-TUBE. After washing with chilled lysis buffer for five times, the pulldown samples were subjected to SDS–PAGE followed by immunoblotting. Input of the GST-Linear-TUBE, GST-TR-TUBE, or GST-control was analyzed by Ponceau S staining.

### *In vitro* protein–protein interaction assays

The method is described elsewhere for ubiquitin chain interaction assays and adapted to GST-dCYLD pulldown [8]. In brief, immobilized GST-fusion proteins were incubated with 500 ng of ubiquitin chains or candidate proteins in chilled binding buffer [50 mM Tris–HCl, pH 7.5, 150 mM NaCl, 0.1% NP-40, 5 mM dithiothreitol (DTT)] at 4°C for 3 h. After washing with the binding buffer for five times, the samples were subjected to SDS–PAGE and immunoblotting.

### *In vitro* ubiquitination assay

Method is described elsewhere [8,30]. Briefly, reaction mixtures containing 100 nM Ube1, 5–20 μM human UbcH7, *Drosophila* UbcD10 or UbcD1/effete, 5–20 μM of the HOIP or LUBEL, and 100 μM ubiquitin in a reaction buffer (50 mM HEPES pH 7.5, 150 mM NaCl, 20 mM $MgCl_2$ in the presence or absence of 1 mM ATP) were incubated at 24°C for 20 min or at indicated time intervals. To stop the reaction, SDS-loading buffer was added and the samples were boiled at 95°C for 1 min. Synthesized chains were resolved in SDS–PAGE, transferred on 0.2-μm nitrocellulose membrane (GE Healthcare) and subjected to immunoblotting.

### Thioester formation assay

The assay was modified from Stieglitz *et al* [30]. In 50 mM HEPES pH 7.5, 150 mM NaCl, and 0.5 mM TCEP reaction buffer, 1 μM of Atto647 (Sigma)-labeled ubiquitin was mixed with 2 μM Ube1 (lanes 1,2); 1 mM ATP and 5 mM $MgCl_2$ were added to start the reaction at 25°C. After 10 min, samples were taken (lanes 3, 4) and 10 μM Effete was added. After 10 min, samples were taken (lanes 5,6), and 20 μM LUBEL-RBR-C was added. Samples were taken after 5 min (lanes 7, 8) to detect the thioester, and 15 μM Ub-His_6 was added and further samples (lanes 9, 10) were taken after 15 min. Gels were scanned with Amersham Imager 600 (GE Healthcare).

### *In vitro* deubiquitination assay

Deubiquitination assays using purified proteins were performed as reported [61]. DUBs were diluted to 2× final concentration (3 μM vOTU and 10 μM OTULIN) in 150 mM NaCl, 25 mM Tris, pH 7.5, and 10 mM DTT and activated at 24°C for 10 min. Subsequently, 10 μl of diluted enzyme was mixed with 1 μg di-ubiquitin and 2 μl of 10× DUB reaction buffer (500 mM NaCl, 500 mM Tris, pH 7.5, and 50 mM DTT) in 20 μl of total reaction mixture. Reaction mixtures were incubated at 37°C, and the reaction was stopped by addition of 2× sample buffer containing β-ME at indicated times.

Ubiquitin cleavage was detected by SDS–PAGE followed by Coomassie-based gel staining using *InstantBlue* (Expedeon, Inc., San Diego, CA). To perform deubiquitination assay on pulldown samples, 2 μg of activated DUBs and samples on pulldown matrix were incubated in the DUB reaction buffer for 30 min at 37°C.

### Mass spectrometry

Proteins were separated by Mini-PROTEAN precast gel (Bio-Rad) and detected by *InstantBlue*. The cutout gel bands were then reduced, alkylated, and digested with trypsin. For the measurement, an UltiMate 3000 HPLC RSLC nano system coupled with a Q Exactive Plus MS, equipped with a Proxeon nanospray source was used (all from Thermo Scientific). The peptides were loaded onto a trap column (PepMap C18, 5 mm × 300 μm ID, 5 μm particles, 100 Å pore size) at a flow rate of 25 μl/min using 0.1% TFA as mobile phase. After 10 min, the trap column was switched in line with the analytical column (PepMap C18, 500 mm × 75 μm ID, 2 μm, 100 Å). Peptides were eluted using a flow rate of 230 nl/min, and a binary 2 h gradient, respectively, 165 min. The gradient started with the mobile phases: 98% A (water/formic acid, 99.9/0.1, v/v) and 2% B (water/acetonitrile/formic acid, 19.92/80/0.08, v/v/v) increased to 35% B over 120 min, followed by a gradient in 5 min to 90% B, maintained for 5 min and decreased in 2 min back to the gradient 98% A and 2% B for equilibration at 30°C. The Q Exactive MS was operated in data-dependent mode, using a full scan ($m/z$ range 370–1,650, nominal resolution of 70,000, target value 3E6) followed by MS/MS scans of the 12 most abundant ions. MS/MS spectra were acquired using normalized collision energy of 27%, with isolation width of 2 and the target value was set to 1E5. Precursor ions selected for fragmentation (charge state 2 and higher) were put on a dynamic exclusion list for 10 s. Additionally, the under fill ratio was set to 20% resulting in an intensity threshold of 4E4. The peptide match feature and the exclude isotopes feature were enabled. For peptide identification, the RAW-files were loaded into Proteome Discoverer (version 1.4.0.288, Thermo Scientific). All hereby created MS/MS spectra were searched using the search engine node MSAmanda [49] against the *Drosophila* sequence database called FlyBase (version dmel_all-translation-r6.06, 22,256 sequences; 20,222,850 residues). The following search parameters were used: Beta-methylthiolation on Cys was set as a fixed modification; oxidation on Met, acetylation on Lys, phosphorylation on Ser, Thr and Tyr, deamidation on Asp and Glu and ubiquitination variants GlyGly and LeuArgGlyGly on Lys were set as variable modifications. Monoisotopic masses were searched within unrestricted protein masses for tryptic peptides. The peptide mass tolerance was set to ± 5 ppm and the fragment mass tolerance to ± 0.03 Da. The maximal number of missed cleavages was set to 2. The result was filtered to 1% FDR using Percolator algorithm integrated in Proteome Discoverer. The localization of the sites of variable modifications within the peptides was performed with the tool ptmRS, integrated in Proteome Discoverer and based on phosphoRS [62]. Therefore, a site probability filter of 100% was used as cutoff.

### Fly husbandry and strains

Fly cultures and crosses were carried out on normal growth medium at 25°C. The following fly strains were used in this study: $w^{1118}$ ($w^-$),

*UAS-RNAi CG11321 GD* (#18055) (Vienna *Drosophila* Resource Center, Vienna, Austria) [63], *Rel^{E20}* (#55714), *dCYLD^{B224}* (#16173), *Mi ET1 CG11321* (#22725), and *24B-GAL4* (#1767) (Bloomington *Drosophila* Stock Center, Bloomington, IN). *Mef2-GAL4* (Bloomington: #27390) and *Tub-GAL4* (Bloomington: #30906) strains kept in a heat shock-hid Y background were gift from Jeroen Dobbelaere (MFPL, Vienna, Austria).

**Generation of LUBEL mutant fly lines by CRISPR/Cas9**

CRISPR-mediated genome editing was used to create catalytically dead LUBEL flies as described previously [64]. The *LUBEL CC2690/2693SS* (*CC/SS*) double mutant lines were generated using gRNA target sequence CATACCTGGCCAGCGAGTAT. The *LUBEL delRING2* line was created as an indel frameshift mutation occurred and genomic DNA sequencing indicated the indel leads to a stop codon at amino acid 2719. The following donor oligo sequence was used to introduce the mutations: GTTAAACATTGAATTTGTAAAAGATA CAACATACTGATATCTTTGTATCTGAACATACCTGGCCAGCGAGT ATCGAAATTTGGACTTGGGAGAGTCTATGCCGTTTTGGGCCAAAT GCTCCTGGACACCCTGGGCCTGCAGCTCCGGATCGTTTTCGCGCTT CCATTCC.

**Bioinformatics**

To identify a *Drosophila* HOIP orthologue, we performed a NCBI-BLASTP search with the human HOIP within the *Drosophila melanogaster* proteome and the highly significant hits (e-value < 1e-88) came from the *CG11321* gene [65]. The reciprocal search with LUBEL protein finds HOIP as best hit in the human proteome (1e-92) and confirms the orthologous relationship. There are seven annotated polypeptides for LUBEL in FlyBase. We selected isoE (Flybase ID FBtr0331216) as the representative isoform containing a conserved catalytic region in the C-terminus, which has a length of 2892 amino acids. For a multiple sequence alignment, we selected orthologues from the NCBI nonredundant database, aligned with MAFFT (version 6, L-INS-I method [66], and visualized with Jalview [22]). For the HOIP family, we used the following sequences and accession numbers: *Drosophila melanogaster* (NP_723214.2), *Apis mellifera* (XP_393719.4), *Pediculus humanus* (XP_002430159.1), *Aplysia californica* (XP_005111217.1), *Takifugu rubripes* (XP_003968217.1), *Xenopus laevis* (NP_001090429.1), *Mus musculus* (NP_919327.2), and *Homo sapiens* (NP_060469.4). In the CYLD alignment, we incorporated sequences of *Drosophila melanogaster* (NP_723554.1), *Apis mellifera* (XP_393824.3), *Pediculus humanus* (XP_002431059.1), *Caenorhabditis elegans* (NP_001255045.1), *Aplysia californica* (XP_005094707.1), *Mus musculus* (NP_775545.1), and *Homo sapiens* (NP_056062.1).

**RNA sequencing**

To prepare RNA-Seq samples, total RNA was isolated from young adult flies using TRIzol (Thermo Fisher Scientific), contaminating DNA digested by TURBO DNA-free Kit (Thermo Fisher Scientific), and Bioanalyzer 2100 (Agilent Technologies) was used to determine the quality and quantity of RNA according to the manufacturers' instructions. The library was prepared from these samples by poly (A) enrichment (New England Biolabs, Ipswich, MA). The resulting

fragmented samples were sequenced on a HiSeq2000 SR with a read length of 50 (by VBCF-NGS). The reads were mapped to the *Drosophila melanogaster* dm3 reference genome either with STAR (version 2.4.0d) [67] or TopHat (version 2.0.9) [68]. Reads aligning to rRNA sequences were filtered out prior to mapping. The read counts for each gene were detected using HTSeq (version 0.5.4p3) [69]. The counts were normalized using the TMM normalization from edgeR package in R. Prior to statistical testing, the data were voom transformed, and then, the differential expression between the sample groups was calculated with limma package in R. The functional analyses were performed using the topGO and gage packages in R. For visualization, heat maps were created using in R and fragment alignments were processed using the Integrative Genomics Viewer (IGV_2.3.40 software) [70,71].

**Analysis of mRNA expression**

To detect mRNA expression level, RNA isolation was performed as described above and cDNA was prepared with oligo(dT)$_{18}$ primer or random priming (Thermo Scientific) of RNA using SuperScript II Reverse Transcriptase (Invitrogen) according to the manufacturer's instructions. Standard RT–PCR was performed to detect the following targets using the primers as follows: AttC-F 5′-GCAGAACA CAAGCATCCTAATCG-3′; AttC-R 5′CCAGCGGGATTGGAGGTTAAG 3′; Dipt-F 5′-GCTGCGCAATCGCTTCTACT-3′; Dipt-R 5′-TGGTGGAG TGGGCTTCATG-3′; Dro-F 5′-CGTGAGAACCTTTTCCAATATGATG 3′; Dro-R 5′-TCCCAGGACCACCAGCAT-3′; Rp49-F 5′-GACGCTTCAA GGGACAGTATCTG-3′; Rp49-R 5′-AAACGCGGTTCTGCATGAG-3′; HSP70-F 5′-CCTGCTATTGGAATCGATCTGGGCACC-3′; HSP70-R 5′-GGACTCACCCTTATACTCCACCCC -3′; LUBEL N-terminal region-F 5′-AGGGAACCCTTACCAGAAACA-3′; LUBEL N-terminal region-R 5′-TATTCCGAGGCACTCTCTTCA-3′; LUBEL catalytic region-F 5′-CA TAGCTGACTCGGAAACTCG-3′; LUBEL catalytic region-R 5′-CTGTT TGCAGTCTTGCCTTTC-3′. Ubi-p63E-F 5′-GCTAAGATCCAAGACAA GGAG-3′; Ubi-p63E-R 5′-GCCTGGATTCCTCCACGGAGACGG-3′.

**Stress treatments**

Newly eclosed adult flies were collected in fresh vials containing normal growth medium and aged 2–5 days. For testing heat susceptibility of adult flies, male and/or female flies were placed at $36 \pm 0.5°C$ in a water bath. Every hour, the mortality rate was estimated by counting the number of flies unable to exhibit a sit-up response after tapping. Knockdown experiments were accompanied by a negative control of the isogenic host strain $w^{1118}$. To detect heat-induced ubiquitinated protein aggregates, total fly lysates were analyzed by immunoblotting. To analyze heat-induced expression of HSP70, flies were incubated in a 36°C water bath for 30 min, and recovered at 25°C for 1 h. To monitor the immune response of adult flies, survival assay and bacteria-induced anti-microbial peptide production were measured. Briefly, adult flies were pricked with either Gram-negative *E. coli* DH5α or Gram-positive *Micrococcus luteus* bacteria (American Type Culture Collection (ATCC), Manassas, VA) using a thin needle (Roboz Surgical Instrument, Gaithersburg, Maryland) and incubated at 25 and 29°C, respectively. To obtain survival count, the infected flies were counted at indicated time points, and to monitor the activation of the immune signaling pathways, AMP levels were measured by qPCR as described [72].

## Negative geotaxis assay

Flies were transferred to fresh vials and acclimated to the environment, undisturbed for 10–15 min. The flies were tapped to the bottom of the vials, and the number of flies that climbed above 2 cm by 5 s after the tap was scored. The flies were rested for at least 1 min before the assay was repeated.

## Histology and immunofluorescence

Flies were either directly cryo-preserved by embedding in Tissue-Tek Optimum Cutting Temperature Compound (Sakura Finetek, Torrance, California) or fixed in Carnoy's fixative for 16–18 h. Frozen blocks were sectioned at a thickness of 10 μm and dried. Fixed flies were processed routinely, embedded in paraffin and sectioned at a thickness of 2 μm. Sections were stained by a routine hematoxylin and eosin (H&E) protocol in a Microm HMS 740 automated stainer. For immunostaining, paraffin sections were deparaffinized, rehydrated, and subjected to antigen retrieval by microwave treatment at 750 W in Sodium Citrate buffer, pH 6.0. Blocking was done in 2% BSA in PBST for 1 h followed by incubation with primary antibody for 48 h at 4°C. Anti-beta-actin antibody (1:250, ab8227, Abcam) was used as a primary antibody. Subsequently slides were washed with blocking buffer and incubated with secondary antibody for 24 h at 4°C. The following secondary antibody was used: Goat anti-rabbit IgG-Alexa Fluor 488 (1:500, ab150077, Abcam). Slides were then washed with blocking buffer and mounted using Vectashield Hardset Mounting Medium with DAPI (H-1500, Vector Laboratories, Burlingame, California). Slides were scanned with a *Pannoramic 250 Flash II* Scanner (3D Histech). Digital slides were reviewed and images acquired with the *Pannoramic Viewer* software (3D Histech).

## Fly longevity assay

For longevity assay, female flies were maintained at 25°C in single-sex groups of no more than 20 flies per group and scored every 2–3 days. Flies were flipped to new vials every 3–4 days. No anesthesia was used during the longevity experiments, numerical ID was given to each vial and vial positions were randomized to minimize variation in experimental design and environment. Data from multiple vials were combined, plotted as survival curves. Total numbers are indicated in the figure legend.

## Statistical analysis

All graphs were created using GraphPad Prism 6 software (GraphPad Software, Inc., La Jolla, CA). Unpaired *t*-test was used to compare two groups and median survival time and log-rank (Mantel–Cox) and Gehan–Breslow–Wilcoxon tests were performed for curve comparison analysis of the fly assays. For RNA expression analysis, RNA-Seq data were analyzed using a two-way analysis of variance (ANOVA) multiple comparison and qPCR data were analyzed using one-way ANOVA, followed by Tukey's post hoc test. Significance and confidence level was set at 0.05.

**Expanded View** for this article is available online.

## Acknowledgements

We acknowledge Conchi Martinez and Ryoko Shiraishi (IMBA, Vienna, Austria), Joseph Gokcezade (IMBA Fly house), Richard Imre (Protein chemistry, IMP-IMBA core facilities), Tamara Engelmaier and Agnieszka Piszczek (HistoPathology, The Vienna Biocenter Core Facilities GmbH (VBCF), Vienna, Austria), Peggy Stolt-Bergner, Katharina Radakovics, and Jana Neuhold (Protein Technologies, VBCF) for their technical supports. We also thank Molecular Biology Service and Biooptics (IMP-IMBA core facilities) and Next-Generation Sequencing Core Facility (VBCF) for the sample processing and analysis, Masayuki Miura (Tokyo Univ., Tokyo, Japan) and Francois Bonnay (IMBA, Vienna, Austria) for constructive discussions and advices. We thank all the members in Ikeda Lab for scientific discussions on the project and for comments on the manuscript, and Life Science Editors for editorial assistance. The work was supported by the ERC consolidator grant (LUbi, 614711), OeNB, and COST (European Cooperation in Science and Technology, PROTEOSTASIS BM1307) (F.I.), Austrian Academy of Sciences (F.I. and L.R), the Francis Crick Institute (grant number FCI01) which receives its core funding from Cancer Research UK, the UK Medical Research Council, and the Wellcome Trust (K.R.), SFB670 DFG (K.H.), the Austrian Science Fund (SFB F3402; P2465-B24; and TRP 308-N15), and MEIOsys (222883-2) (K. M.).

## Author contributions

TA, JA, CE, IT, LD, and PD performed experiments. AS, SJ, AG, and KH performed bioinformatic analysis. LR, KM, and AK analyzed data. FI and KR coordinated the study and wrote the manuscript. All the authors discussed the results and commented on the manuscript.

## Conflict of interest

The authors declare that they have no conflict of interest.

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
