## [Review Process File · EMBO Reports]

Manuscript EMBO-2016-42378

Linear ubiquitination by LUBEL plays a critical role in *Drosophila* heat stress response

Tomoko Asaoka, Jorge Almargo, Christine Ehrhardt, Alexander Schleiffer, Sini Junttila, Leonie Ringrose, Karl Mechtler, Anoop Kavirayani, Attila Gyenesei, Kay Hofmann, Peter Duchek, Katrin Rittinger, and Fumiyo Ikeda

Corresponding author: Fumiyo Ikeda, Institute of Molecular Biotechnology (IMBA)

Review timeline:	Submission date:	15 March 2016
	Editorial Decision:	19 April 2016
	Revision received:	29 July 2016
	Editorial Decision:	24 August 2016
	Revision received:	30 August 2016
	Accepted:	05 September 2016

Editor: Achim Breiling¹

Transaction Report:

1st Editorial Decision

19 April 2016

Thank you for the submission of your research manuscript to EMBO reports. We have now received the full set of referee reports that is copied below.

As you will see, all three referees acknowledge the potential interest of the findings. However, all three referees have raised a number of concerns and suggestions to improve the manuscript or to strengthen the data. In particular referee #3 has several comments that need to be addressed. Most importantly, the concerns on the involvement of linear ubiquitination in heat stress response (referee #4) should be addressed.

Given these constructive comments, we would like to invite you to revise your manuscript with the understanding that all referee concerns (as detailed in their reports) must be fully addressed in a complete point-by-point response. Acceptance of the manuscript will depend on a positive outcome of a second round of review. It is EMBO reports policy to allow a single round of revision only and acceptance or rejection of the manuscript will therefore depend on the completeness of your responses included in the next, final version of the manuscript.

Revised manuscripts should be submitted within three months of a request for revision; they will otherwise be treated as new submissions. Please contact us if a 3-months time frame is not sufficient for the revisions so that we can discuss the revisions further.

Supplementary/additional data: The Expanded View format, which will be displayed in the main

HTML of the paper in a collapsible format, has replaced the Supplementary information. You can submit up to 5 images as Expanded View. Please follow the nomenclature Figure EV1, Figure EV2 etc. The figure legend for these should be included in the main manuscript document file in a section called Expanded View Figure Legends after the main Figure Legends section. Additional Supplementary material should be supplied as a single pdf labeled Appendix. The Appendix includes a table of content on the first page, all figures and their legends. Please follow the nomenclature Appendix Figure Sx throughout the text and also label the figures according to this nomenclature. For more details please refer to our guide to authors.

Important: All materials and methods should be included in the main manuscript file.

Regarding data quantification and statistics, can you please specify the number "n" for how many experiments were performed, the bars and error bars (e.g. SEM, SD) and the test used to calculate p-values in the respective figure legends? This information must be provided in the figure legends. Please also include scale bars in all microscopy images.

I look forward to seeing a revised version of your manuscript when it is ready. Please let me know if you have questions or comments regarding the revision.

REFEREE REPORTS

Referee #2:

This manuscript describes the identification of the *Drosophila* HOIP homolog, its characterization *in vitro* and *in vivo* and a first analysis of its function in the fly. Intriguingly in the fly HOIP is not found as part of LUBAC complex, but forms a much larger protein itself. The flies also have no obvious homolog of the DUB for linear chains, Otulin, but the dCYLD can degrade linear chains.

In line with these differences the function of HOIP in flies also is different, since it seems to play no role in immune response. However, loss of HOIP function results in clear phenotypes upon heat stress, and particularly in muscles.

The experiments are very clear and thorough and the paper is very well written. Still some points need to be clarified and/or worked out for publication

1) It's not clear how a catalytic cys to ala mutant can still have linear chain forming activity. It suggests that this is not the (only) catalytic cysteine and suggests in fact that a different cysteine does the job and that this cys has a supportive role.

This analysis (fig 2) was performed with 'recombinant' protein, but it is not clear to me whether this was all done with *E. coli* expressed protein, or whether some of this was expressed in insect cells. In the latter case it is of course possible that some endogenous protein was copurified.

2) The statement 'UAS-shLUBEL crossed with Mef2-Gal4 flies significantly decreased survival upon heat shock' seems too simplistic. In fact, as stated elsewhere in the manuscript, the median survival is not significantly different from WT. Nevertheless, there is a phenotype, maybe something that causes an initial problem after heat shock in a subset of flies, but over time can be recovered from? This needs to be clarified.

Minor points

- It seems a pity to give this protein a separate name, as its enzymatic function and sequence do resemble HOIP most.

- Figure 2 C labels are misaligned with the lanes, which makes it hard to read

Referee #3:

Linear ubiquitin chains are known to be involved in NF-kappaB activation induced by various stimuli including NOD2 signaling and protection cells from death in mammals. LUBAC (linear ubiquitin assembly complex), of which HOIP is the catalytic center, is the only identified E3 to

generate linear ubiquitin chain specifically. In this manuscript, Asaoka et al. proposed that C-terminal part of *Drosophila* CG11321, which is homologous to the catalytic portion of HOIP (RING-IBR-RING-LDD), can support formation of linear ubiquitin chains *in vitro* and then they named CG11321 as LUBEL (linear ubiquitin E3 ligase). The authors also showed that catalytic activity of CG11321 is involved in heat shock responses in the organism. The reviewer agrees with the authors that C-terminal part of CG11321 supports generation of linear ubiquitin chains *in vitro*. However, not only biochemical but also physiological characterization of CG11321 is not enough to support the authors' conclusion. Thus, the reviewer feels any enthusiasm to include this manuscript in the EMBO report at least in the present form.

Major points

#1. HOIP has two binding partners, HOIL-1 and SHARPIN. The N-terminal region of HOIP provides binding sites to the two associate subunits and is regarded to be the auto-inhibitory region. The authors showed that the UBA1 domain of CG11321 is homologous to the UBA of HOIP, which is the binding site of HOIL-1. Thus, it seems plausible that CG11321 has some binding partners and its ubiquitin ligase activity is suppressed without the partner(s). It is important to show that CG11321 has binding partner(s) or not. Also it is of great importance to show whether full length of CG11321 with binding partners (if they exist) has the activity to generate linear ubiquitin chains.

#2. The authors showed that UBA2 and NZF of CG11321 have ubiquitin binding activity in Figure 1D. Both domains did interact with ubiquitin chains longer than tetramers. The reviewer is afraid that the ubiquitin binding activity of these domains are very weak. The binding affinity to each ubiquitin chain using di-ubiquitin should be evaluated quantitatively by SPR or ITC.

#3. The RING-IBR-RING E3s forms a thio-ester bond with ubiquitin at the specific Cys residue in RING2 and C2704 is the specific Cys in RING2. The C2704A mutant of C-terminal part of CG11321 exhibits activity to generate linear ubiquitin chains although the activity of the mutant is weaker than that of WT. Thus, the catalytic mechanism of CG11321 may be different from that of HOIP. The authors should examine CG11321 forms a thio-ester bond with ubiquitin at C2704.

#4. It has been shown that the C-terminal part of HOIP called as the LDD domain is critical for the generation of linear chains and critical amino acid residues for linear ubiquitination are known. It is worth trying to examine whether mutations of the critical residues in LDD of CG11321 affects the linear chain generation by C-terminal part of CG11321.

#5. The polyubiquitin (linear ubiquitin) gene is encoded in *Drosophila* genome and expression of the polyubiquitin gene is heat inducible. It is of great importance to examine whether increase of linear ubiquitin in heat shock is derived from the polyubiquitin gene or generated from ubiquitin monomers.

#6. No ubiquitin signal could be detected in GST-linear-TUBE pulled down materials treated with vOTU in Figure 3A although linear ubiquitin signal can be detected. It is very curious why so abundant ubiquitinated (non-linear) materials are co-purified with GST-linear-TUBE. The results of Figure 3A indicated that the amount of linear ubiquitin in fly is very small. The linear ubiquitin detected by GST-linear TUBE may be the products of the polyubiquitin gene and loss of E3 activity of CG11321 may suppress the transcription of the polyubiquitin gene.

#7. The authors observed that the amount of linear ubiquitin in cells were increased in cells expressing dCYLD C284S in Figure 4DE, Does introduction of dCYLD C284S increase the amount of K63 chains as well?

#8. Does dCYLD interact with CG11321?

#9. The life span of CG11321 mutant flies is shorter than WT (Figure 5C). It is very curious whether the authors found any abnormality in muscles of aged mutant flies.

#10. Blots are not clear enough to support the conclusion in Figure 5B.

Referee #4:

1. What are the major claims and how significant are they

Linear ubiquitinations are well documented in mammals, where they contribute to a variety of mechanisms, including the inflammatory response. Here the authors describe linear ubiquitination in insect cells. They describe LUBEL (CG11321) as the unique linear ubiquitination enzyme and identify dCYLD as a De-ubiquinating (DUB) enzyme regulating K63 & linear-ubiquitination. The authors claim that LUBEL plays a role in *Drosophila* muscles during the heat stress response.

2. Are the claims novel and convincing?

The identification in *Drosophila* of linear ubiquitin chains as well as characterization of LUBEL and CYLD are convincing, although this is not a novel finding as similar mechanisms are already existing in mammals and are very well documented. Additionally the claim that LUBEL and linear ubiquitin chain formation are required *in vivo* in muscles to survive the heat stress response is not supported by the author's data (see below).

3. Are the claims appropriately discussed in the context of earlier literature?

4. Who will be interested and why?

The paper may interest readers working in the fields of DUBs/ubiquitination or the heat stress response. As flies deprived of linear-ubiquitin chains (the mutants analyzed here by the authors) are morphologically normal, viable, fertile and of normal longevity, the identification of linear-ubiquitin chains in insects is of limited interest.

5. Does the paper stand out in some way from the others in its field?

The paper is like many others in the field describing the enzymatic processes leading to linear ubiquitin chain formation (i. e. identification of the enzymes). However, the function of these linear ubiquitin chains in insects remains obscure.

6. Are the experimental data of sufficient quality to justify the conclusions?

In my opinion, the conclusions are well supported by the data, except for the heat stress response.

More precisely, the authors state that linear and K48 ubiquitinations are increased upon high temperature exposure, although in fig7A, formation of linear ubiquitin chains seems unchanged whereas K48 and k63 chains are clearly increased. The authors should modify the text and conclusions or provide additional/new data to support their claim. The survival experiment in fig7C does not support the conclusion of a role for LUBEL in muscle heat-stress response as *MeF2-Gal4/Lubel* flies die at a similar rate as *MeF2/Gal4/+* control flies.

The authors show a slight reduction of Hsp70 expression upon heat stress response in flies deprived of functional LUBEL for a single time point. Is this true over time, or is there only a slight delay in HSP70 gene expression?

Additionally, the RNaseq data presented in fig6C are not really comprehensible and need reformatting for a better understanding of the figure.

1st Revision - authors' response

29 July 2016

Thank you for your interest in our manuscript 'Linear ubiquitination by LUBEL plays a critical role in *Drosophila* heat stress response', by Asaoka et al.

We are happy with referees' positive comments and constructive suggestions on our manuscript. We have now addressed the points raised by reviewers with several new experiments, as described in greater detail in the point-by-point response. Most importantly, we clarified that the linear ubiquitin chain accumulation induced by heat shock is not due to ubiquitin-gene induction. In addition, we demonstrate that LUBEL catalytic mutant flies have defective linear ubiquitination and die rapidly upon heat shock.

In short, in the revised manuscript, we included new data that show;

- 1) LUBEL ubiquitin E3 ligase catalyzes linear ubiquitin chains by a RING/HECT hybrid mechanism, (a new figure, Fig 2C)
- 2) Unlike a human homologue HOIP, recombinant full-length LUBEL without potential binding partners is sufficient to generate linear ubiquitin chains, (new figures Fig 2E and Fig EV2G)
- 3) LUBEL interacts with a deubiquitinase dCYLD, (new figures Fig 3C and D)
- 4) Induction of linear ubiquitin chains by heat shock is independent from ubiquitin gene induction, (a new figure Fig 5B)

- 5) Muscle-specific LUBEL knockdown in flies leads to a significantly reduced survival rate at an early time point after heat shock, which matches the defective HSP70 induction at an early time point (new figures, Fig EV5E, F and modified Fig 6C)
- 6) Muscle morphology is unaffected in LUBEL mutant flies, determined by histological analysis. (new figures Fig 5D and Appendix Fig S2)

To make the major changes in the revised manuscript clear, we have indicated them in red. These new data have strengthened the conceptual advance provided by our work. Our findings move the ubiquitination field forward substantially by uncovering an E3 ligase that is sufficient to catalyze linear ubiquitination and revealing a novel role for this modification in the fly heat shock response. Further, these mutant flies provide a tool to further dissect the physiological relevance of linear ubiquitination *in vivo*. We are confident that the revised manuscript meets the criteria and is acceptable for publication in the EMBO Reports.

We look forward to hearing a positive response.

We thank reviewers for their constructive suggestions on our manuscript. Please find below as the point by point response.

Referee #2

This manuscript describes the identification of the *Drosophila* HOIP homolog, its characterization *in vitro* and *in vivo* and a first analysis of its function in the fly. Intriguingly in the fly HOIP is not found as part of LUBAC complex, but forms a much larger protein itself. The flies also have no obvious homolog of the DUB for linear chains, Otulin, but the dCYLD can degrade linear chains. In line with these differences the function of HOIP in flies also is different, since it seems to play no role in immune response. However, loss of HOIP function results in clear phenotypes upon heat stress, and particularly in muscles.

The experiments are very clear and thorough and the paper is very well written. Still some points need to be clarified and/or worked out for publication

We thank this reviewer for their positive comments. Please find our detailed point-by-point response below.

- 1) It's not clear how a catalytic cys to ala mutant can still have linear chain forming activity. It suggests that this is not the (only) catalytic cysteine and suggests in fact that a different cysteine does the job and that this cys has a supportive role.

We agree with the referee that this is an important point. To address this concern, we first performed an *in vitro* ubiquitination assay with human HOIP-RBR-C wt and HOIP-RBR-C C885A (equivalent to LUBEL-RBR-C C2704A) (see right panels) using identical conditions as used in our study. Under the condition we used (15 min reaction), human HOIP-RBR-C C885A also shows minor, but residual activity to form short ubiquitin chains *in vitro*, very similar to LUBEL-RBR-C C2704A shown in Fig 2A. This might be an E3-independent non-specific activity. We want to highlight that the signals we detect by immunoblot using anti-Linear ubiquitin antibody is more sensitive than the ones detected by anti-pan ubiquitin antibody or by Coomassie staining (please see also the figure panels in Fig 2D and Fig 3A).

Importantly, to confirm that LUBEL acts as a RING/HECT hybrid and that C2704 is indeed the active site cysteine, we performed an *in vitro* ubiquitin loading assay (a new Fig. 2C). The C2704A mutation abolished the signal of ubiquitin loading, which further strengthens the conclusion that C2704 is the major ubiquitin thioester-bond loading site. Taken together, these data strongly suggest that the catalytic Cys in LUBEL is C2704 and that it plays a similar role to human HOIP C885.

[Data removed upon author's request]

This analysis (fig 2) was performed with 'recombinant' protein, but it is not clear to me whether this was all done with *E. coli* expressed protein, or whether some of this was expressed in insect cells. In the latter case it is of course possible that some endogenous protein was copurified.

All of the *in vitro* ubiquitination assays using LUBEL-RBR-C and UBA2-RBR-C (wild type or mutants) were performed using recombinant proteins purified from *E. coli*. Therefore, these experiments are not confounded by co-purification of endogenous proteins from insect cells and strongly suggest that LUBEL is sufficient for the formation of linear ubiquitin chains. We clarified this point in the Material and Method section (Protein purification).

2) The statement 'UAS-shLUBEL crossed with Mef2-Gal4 flies significantly decreased survival upon heat shock' seems too simplistic. In fact, as stated elsewhere in the manuscript, the median survival is not significantly different from WT. Nevertheless, there is a phenotype, maybe something that causes an initial problem after heat shock in a subset of flies, but over time can be recovered from? This needs to be clarified.

We thank the reviewer for raising an important point. To further understand the role of LUBEL in muscle in response to heat shock, we employed an additional muscle-specific driver line called *24B-Gal4* (a new Fig EV5F). The knockdown efficiency in *24B-Gal4>LUBEL* knockdown flies is similar to *Tub-Gal4>LUBEL* knockdown flies (a new Fig EV5F). *24B-Gal4>LUBEL* knockdown flies show a significantly reduced survival of heat shock (by the Gehan-Breslow-Wilcoxon test), though the effect is milder than *Tub-Gal4>LUBEL* flies. Similarly, the climbing ability of *24B-Gal4>LUBAC* flies upon heat clearly goes down upon heat shock (a new Movie EV5).

The effect at the early time point of heat shock is similar in *Mef2-Gal4>LUBEL* flies, suggesting that LUBEL in muscle is important in particular at an early time point. At the later time point, there is a possibility that LUBEL in other tissues may play a role. To explore this possibility, we attempted to generate neuron-specific LUBEL knockdown flies by using *Elav-Gal4* line and analyze the effect of heat shock (see below). The survival rate of the control *Elav-Gal4/+* flies and *Elav-Gal4>LUBEL* upon heat shock did not show any significant differences. This may be due to 1) inefficient knockdown of LUBEL in neurons, which we could not determine due to lack of an antibody that is suited for the immunohistochemistry, or 2) LUBEL in other tissues from neurons and muscle plays an additional role in the later time point of heat shock.

In conclusion, muscle specific knockdown of LUBEL in flies significantly affects the heat shock-dependent survival at an early time point, and not at a late time point. We discussed this more clearly in the revised manuscript.

Minor points

- It seems a pity to give this protein a separate name, as its enzymatic function and sequence do resemble HOIP most.

As the reviewer pointed out, it is indeed a pity that we cannot use the gene name *HoiP* for *CG11321*. This is due to an existing *Drosophila* gene named *hoip* (*CG3949*) encoding a different gene as *LUBEL*. Furthermore, HOIP is short for HOIL-1-interacting protein, and since we could not identify a *Drosophila* homologue of HOIL-1, we have decided to give an alternative name.

- Figure 2 C labels are misaligned with the lanes, which makes it hard to read

We changed the labelling to make it clearer to understand (now in Fig 2B).

Referee #3:

Linear ubiquitin chains are known to be involved in NF- κ B activation induced by various stimuli including NOD2 signaling and protection cells from death in mammals. LUBAC (linear ubiquitin assembly complex), of which HOIP is the catalytic center, is the only identified E3 to generate linear ubiquitin chain specifically. In this manuscript, Asaoka et al. proposed that C-terminal part of *Drosophila* CG11321, which is homologous to the catalytic portion of HOIP (RING-IBR-RING-LDD), can support formation of linear ubiquitin chains *in vitro* and then they named CG11321 as LUBEL (linear ubiquitin E3 ligase). The authors also showed that catalytic activity of CG11321 is involved in heat shock responses in the organism. The reviewer agrees with the authors that C-terminal part of CG11321 supports generation of linear ubiquitin chains *in vitro*. However, not only biochemical but also physiological characterization of CG11321 is not enough to support the authors' conclusion. Thus, the reviewer feels any enthusiasm to include this manuscript in the EMBO report at least in the present form.

We thank the reviewer for their input. Please find below our point-by-point responses.

Major points #1. HOIP has two binding partners, HOIL-1 and SHARPIN. The N-terminal region of HOIP provides binding sites to the two associate subunits and is regarded to be the auto-inhibitory region. The authors showed that the UBA1 domain of CG11321 is homologous to the UBA of HOIP, which is the binding site of HOIL-1. Thus, it seems plausible that CG11321 has some binding partners and its ubiquitin ligase activity is suppressed without the partner(s). It is important to show that CG11321 has binding partner(s) or not. Also it is of great importance to show whether full length of CG11321 with binding partners (if they exist) has the activity to generate linear ubiquitin chains.

We agree with the reviewer that the possibility of additional binding partners of LUBEL is an interesting point. To test if full-length LUBEL does need a potential binding partner to support catalytic activity, as has been observed with mammalian HOIP, we examined if full-length LUBEL is sufficient for linear ubiquitination. Very interestingly, although transiently-expressed, full-length human HOIP requires co-expression of HOIP-1L to produce linear ubiquitin chains in HEK293T cells, full-length LUBEL is clearly capable of catalyzing linear ubiquitin chain formation in insect S2 cells without co-transfection of a potential interacting partner (**added as a new Fig 2E**). We are aware that this experiment cannot exclude the possibility that endogenously expressed proteins in S2 cells may interact with exogenously-introduced LUBEL, and that this putative interaction is important. However, transient expression of HOIP alone is not sufficient to generate linear ubiquitin chains despite the endogenous expression of Sharpin and HOIL-1L in HEK293T cells, strongly suggesting that LUBEL can generate linear ubiquitin chains without partner proteins. To exclude the possibility of an interaction with endogenous S2 proteins, we purified recombinant full-length LUBEL using an insect expression system with Baculovirus. Unlike human HOIP, recombinant LUBEL generates linear ubiquitin formation *in vitro* (**a new Fig EV2G**), further supporting that full-length LUBEL is sufficient to generate linear ubiquitin chains without partner proteins.

As pointed out by the reviewer, we also wondered if the LUBEL-UBA1 domain may mediate protein-protein interactions and recognize UBL-containing *Drosophila* proteins. We aimed to identify the UBL-containing *Drosophila* proteins by bioinformatics. However, the homology of the UBL domain in *Drosophila* proteins to HOIL-1L UBL is rather low. In summary, these observations provide strong support that full-length LUBEL does not necessarily require binding partners for linear ubiquitination. This is now discussed in the "Discussion" section. Given that full-length LUBEL is sufficient for linear ubiquitination without additional partners, we feel that the identification of interacting partners goes beyond the scope of this study.

#2. The authors showed that UBA2 and NZF of CG11321 have ubiquitin binding activity in Figure 1D. Both domains did interact with ubiquitin chains longer than tetramers. The reviewer is afraid that the ubiquitin binding activity of these domains are very weak. The binding affinity to each ubiquitin chain using di-ubiquitin should be evaluated quantitatively by SPR or ITC.

This is a good point and we are grateful to this reviewer to make us reconsider our experiments, as we have now realized that the GST-UBA2 AAA mutant is very unstable and that the majority of protein used in the pulldown studies is degraded. For this reason, we have removed these data and have repeated the pulldown studies with the NZF domain and WT UBA2 domain using linear, K48- and K63-di-ubiquitin (a new Fig 1B). These new experiments showed that the UBA2 domain has a preference for binding to K63 di-ubiquitin. As requested we have also quantified the interaction between the UBA2 domain and different di-ubiquitin chains by ITC. We could not detect any interaction under the experimental conditions ($\sim 50 \mu\text{M}$ UBA2 domain titrated with $\sim 500 \mu\text{M}$ mono or di-ubiquitin) between the UBA2 domain and mono, linear or K48-linked di-ubiquitin. In contrast, there was a clear interaction between UBA2 and K63 di-ubiquitin, which occurred with an affinity of around $20 \mu\text{M}$. However, as can be seen from the ITC trace there is a lot of background signal and not a lot of curvature in the isotherm (due to the relatively low K_d) and hence we would prefer not to add the titrations to the manuscript as we do not think that the actual affinity is relevant for this story.

#3. The RING-IBR-RING E3s forms a thio-ester bond with ubiquitin at the specific Cys residue in RING2 and C2704 is the specific Cys in RING2. The C2704A mutant of C-terminal part of CG11321 exhibits activity to generate linear ubiquitin chains although the activity of the mutant is weaker than that of WT. Thus, the catalytic mechanism of CG11321 may be different from that of HOIP. The authors should examine CG11321 forms a thio-ester bond with ubiquitin at C2704.

To explore if LUBEL C2704 forms a thioester intermediate with ubiquitin we performed a thioester formation assay (according to Stieglitz et al., EMBO reports (2012)) (a new Fig 2C). Similar to human HOIP, WT LUBEL forms a thioester intermediate with ubiquitin, whereas the LUBEL C2704A mutant is no longer able to form the intermediate, identical to HOIP C885A (Stieglitz et al., EMBO reports (2012)). We believe that the very low residual activity observed with the C2704A mutant (Fig 2A) is due to a very low level of E3 independent non-specific activity.

#4. It has been shown that the C-terminal part of HOIP called as the LDD domain is critical for the generation of linear chains and critical amino acid residues for linear ubiquitination are known. It is worth trying to examine whether mutations of the critical residues in LDD of CG11321 affects the linear chain generation by C-terminal part of CG11321.

As suggested, we investigated if the predicted LDD region in LUBEL also plays an important role in binding the acceptor ubiquitin during ubiquitin chain synthesis as it does in HOIP. This is a good point, as the amino acid sequence itself is highly conserved between LUBEL-LDD and HOIP-LDD (labelling for the LDD region is added in Fig EV1B). As previously shown, HOIP R935 and HOIP D936 in LDD are critical for the linear ubiquitin chain formation (Stieglitz et al., Nature (2013)). We performed *in vitro* ubiquitination assay using LUBEL-WT, R2754A (equivalent to HOIP R935A), D2755A (equivalent to HOIP D936A), and C2704A (catalytically dead). Similar to the human HOIP-LDD mutants, mutations in the LDD reduce the ability of LUBEL to synthesize linear ubiquitin chains (a new Fig 2D). Interestingly, the effect of the D2755A mutant is less pronounced than of R2754A. Structural studies will be required to fully explain these subtle differences. In summary, LUBEL-LDD is important for generating linear ubiquitin chains.

#5. The polyubiquitin (linear ubiquitin) gene is encoded in Drosophila genome and expression of the polyubiquitin gene is heat inducible. It is of great importance to examine whether the increase of

linear ubiquitin in heat shock is derived from the polyubiquitin gene or generated from ubiquitin monomers.

This is a critical point. In *Drosophila*, *Ubi-p63E* and *Ubi-p5E* encode tandem-repeat ubiquitin polymers (10 repeats and 7 repeats, respectively). We attempted to examine heat-dependent inductions of these by qPCR and RNA seq using heat-induced fly samples, though we expected to be technically challenging due to the repetition of the nucleotide sequences. Based on qPCR, we found that *Ubi-p63E* is induced by heat treatment as previously shown (Niedzwiecki and Fleming., 1993) (added a new Fig EV5B). Importantly, the heat-induced expression of *Ubi-p63E* is not affected by mutations in LUBEL, suggesting that the mutant phenotypes are not a consequence of altered *Ubi-p63E* expression. For *Ubi-p5E*, we tried with 3 different primer sets including the one used in a recent publication (Ristic et al., JBC, 2016), however, we were not successful in amplifying a specific fragment by PCR (see below, lane 3-5). Therefore, we could not validate these samples by qPCR, while sample #1 and #6 was used for the qPCR validation.

Due to the sequence repetitions, RNA seq analysis did not provide reliable results for *Ubi-p63E* and *Ubi-p5E*, as many of the sequence reads were not annotated with a specific region of the genes. On the other hand, at the protein level, we found that linear ubiquitin chains in various lengths are induced upon heat shock. The expected molecular weights of the unprocessed gene products are 85.8 kDa (*Ubi-p63E*) and 60.0 kDa (*Ubi-p5E*), however, we observed that heat-induced linear Ub chains correspond to different sizes (Fig 6A, please also see a response to the referee #4, point 6). These observations collectively suggest that the linear ubiquitin chains induced by heat shock in flies is at least partly independent from the gene induction.

#6. No ubiquitin signal could be detected in GST-linear-TUBE pulled down materials treated with vOTU in Figure 3A although linear ubiquitin signal can be detected. It is very curious why so abundant ubiquitinated (non-linear) materials are co-purified with GST-linear-TUBE.

Indeed, this is an interesting point. Because the purpose of linear-TUBE pulldown for Fig 3A was to enrich the total linear Ub population, we did not use denatured protein extracts for the experiments. Therefore, we expected that GST-linear-TUBE pulls down ubiquitinated proteins (not necessarily linear ubiquitination), which are in a complex with linear ubiquitin chains, linearly ubiquitinated proteins, or ubiquitinated proteins modified with mix-linkage types of chains. This is indeed the case (Fig 3A). This point is discussed in the Discussion section.

The results of Figure 3A indicated that the amount of linear ubiquitin in the fly is very small. The linear ubiquitin detected by GST-linear TUBE may be the products of the polyubiquitin gene and loss of E3 activity of CG11321 may suppress the transcription of the polyubiquitin gene.

Similar to the above (point #5), we confirmed that LUBEL mutation does not affect the transcripts of *Ubi-p63E* at the basal condition by qPCR (see below). In addition, the expected molecular weights of unprocessed forms of *Ubi-p63E* and *Ubi-p5E* gene products are 85.8 and 60.0 kDa, however, we observed that linear Ub chains with various sizes and different from 85.8 and 60.0 kDa (Fig 3A). These observations collectively indicate that indeed the total amount of linear

ubiquitination is only detected at low levels, and LUBEL catalytic activity is not involved in gene regulation of *Ubi-p63E*.

#7. The authors observed that the amount of linear ubiquitin in cells were increased in cells expressing dCYLD C284S in Figure 4DE, Does introduction of dCYLD C284S increase the amount of K63 chains as well?

This is also an interesting point. To investigate whether dCYLD C284S has any effects on the K63 chain protection, we generated a new tool, immobilized GST-pan Ub-TUBE, which is a linkage type-independent ubiquitin chain enrichment matrix, called TR-TUBE (Yoshida et al., 2015). By using GST-TR-TUBE, we observed an increase in K63-linked ubiquitin chains in dCYLD mutant adult flies in comparison to the control flies (**added as a new Fig EV3B**). On the other hand, in S2 cells, transient expression of dCYLD C284S mutant did not have any clear dominant negative effect based on the unchanged amount of K63-linked ubiquitin chains (**see right**). This observation suggests that there are possibly potent DUBs against K63-linked chains in particular in S2 cells, which dCYLD C284S mutant cannot compete with.

#8. Does dCYLD interact with CG11321?

This is in particular an interesting point as the human HOIP-PUB domain, which has been suggested to be involved in recruitment of human CYLD, is not predicted in LUBEL. To examine the interaction between LUBEL and dCYLD, we performed a GST-pulldown assay using GST-dCYLD, and TCL of S2 cells transfected with LUBEL-RBR-C or recombinant LUBEL-RBR-C (**in a new Fig EV3C and D**). Unexpectedly, we detected an apparently direct interaction (**a new Fig EV3D**), although in the mammalian system the interaction occurs via SPATA2 (Wagner et al., 2016, EMBO J), suggesting that the mechanism of LUBEL/CYLD interplay is different in fly. However, there is a CYLD homologue in *Drosophila* called Tamo (Minakhina et al., 2003, Genes Cells), which is suggested to be a negative regulator of NF- κ B/Rel pathway. Characterization of these proteins and the interplay of Tamo, LUBEL and dCYLD in *Drosophila* need further studies. We also discussed about this point in the Discussion section.

#9. The life span of CG11321 mutant flies is shorter than WT (Figure 5C). It is very curious whether the authors found any abnormality in muscles of aged mutant flies.

To address if aged flies have any abnormalities in muscle functions, we analyzed aged LUBEL mutant flies (50-day old males or 60-day old females). These flies displayed no obvious locomotion defects when compared to the control flies by climbing assay (**new Movies EV2 and 3**). At the tissue morphological level, we found that the muscle tissues of aged flies (control, LUBEL-CC/SS

and LUBEL-delR2) are not affected based on the histological analysis of H&E and Actin staining (new Fig 5D, Appendix S1A and B).

#10. Blots are not clear enough to support the conclusion in Figure 5B.

We repeated the experiments and now have clearer data. The results were exchanged to the new data set (exchanged in Fig 5B).

Referee #4:

1. What are the major claims and how significant are they Linear ubiquitinations are well documented in mammals, where they contribute to a variety of mechanisms, including the inflammatory response. Here the authors describe linear ubiquitination in insect cells. They describe LUBEL (CG11321) as the unique linear ubiquitination enzyme and identify dCYLD as a De-ubiquinating (DUB) enzyme regulating K63 & linear-ubiquitination. The authors claim that LUBEL plays a role in Drosophila muscles during the heat stress response.

We thank the reviewer for their thoughtful comments. Please find below our point-by-point response.

2. Are the claims novel and convincing? The identification in Drosophila of linear ubiquitin chains as well as characterization of LUBEL and CYLD are convincing, although this is not a novel finding as similar mechanisms are already existing in mammals and are very well documented.

We are pleased that the reviewer found our data convincing. With the additional new data in the revised manuscript, we would like to emphasize that our findings are indeed novel, especially in terms of 1) LUBEL's ability to generate linear ubiquitin chains **without** Sharpin or HOIL-1L (or similar UBL-containing binding partners) (see new Fig 2E and Fig EV2G), 2) heat-shock induces linear ubiquitin chains in flies (Fig 6A), 3) a catalytically dead mutant of LUBEL is sufficient to de-regulate heat-shock responses in flies (Fig 6B). While we and others have previously shown the functions of linear ubiquitin chains in the regulation of immune responses and cell death in mammals, we consider that the above mentioned points are novel and highlight an unexpected diversity in the function of linear ubiquitin chains in animals and the mechanism of the E3 ligase synthesizing them.

Additionally the claim that LUBEL and linear ubiquitin chain formation are required in vivo in muscles to survive the heat stress response is not supported by the author's data (see below).

We tried to address this point as detailed below (please see point #6).

3. Are the claims appropriately discussed in the context of earlier literature?

4. Who will be interested and why? The paper may interest readers working in the fields of DUBs/ubiquitination or the heat stress response. As flies deprived of linear-ubiquitin chains (the mutants analyzed here by the authors) are morphologically normal, viable, fertile and of normal longevity, the identification of linear- ubiquitin chains in insects is of limited interest.

We find that the identification of linear ubiquitin in insects is important because:

-It highlights a novel role of linear ubiquitin which could even be important in mammals.
-It is an interesting new model system where loss of linear chains can be studied in a whole organism.

We hope that the reviewer agrees with us on this aspect.

5. Does the paper stand out in some way from the others in its field? The paper is like many others in the field describing the enzymatic processes leading to the linear ubiquitin chain formation (I. e. identification of the enzymes). However, the function of these linear ubiquitin chains in insects remains obscure.

We aimed to address the functions of linear ubiquitin chains in flies by performing additional experiments as described below (Point #6).

6. Are the experimental data of sufficient quality to justify the conclusions? In my opinion, the conclusions are well supported by the data, except for the heat stress response. More precisely, the authors state that linear and K48 ubiquitinations are increased upon high temperature exposure, although in fig7A, formation of linear ubiquitin chains seems unchanged whereas K48 and k63 chains are clearly increased. The authors should modify the text and conclusions or provide additional/new data to support their claim.

We thank the reviewer for pointing out a critical issue. We observed an increase of total, K48 and linear chains, but not K63 as in the original figure panels. We think that the format in the previous figures was rather confusing, therefore we changed the style and clarified the points also in the text (as Fig 6A and Fig EV5A). We repeated experiments as below confirming that heat induces linear ubiquitination signal in control (*w*⁻) flies (below, left panels). In addition, we compared linear ubiquitination signal in control (*w*⁻) and LUBEL mutant CC/SS flies. Though there are some background signal, heat-dependent induction of high-molecular weight linear ubiquitin chains was only observed in control (*w*⁻) flies (see below A) and not in LUBEL mutant CC/SS flies (see below B).

Collectively, these results suggest that linear ubiquitination is induced upon heat shock in flies.

The survival experiment in fig7C does not support the conclusion of a role for LUBEL in muscle heat-stress response as *Mef2-Gal4/Lubel* flies die at a similar rate as *Mef2/Gal4*⁺ control flies.

This is indeed a critical point for this manuscript, and similar to the point by Referee #2, point #2 (**Please also see response to Referee #2**). We further analyzed the possible functions of LUBEL in muscle by using one additional muscle-specific Gal4 line (*24B-Gal4*) and confirmed that the phenotype is similar to *Mef2*-dependent LUBEL knockdown (**a new Fig EV5F**). Importantly, we did observe significant differences in the heat-dependent survival rate between control and muscle-specific LUBEL knockdown lines at early time points (**a new Fig EV5F and modified Fig 6C**). This has been clarified also in the text.

These observations indicate that the heat shock response in muscle-specific LUBEL knockdown flies is affected especially at early time points, whereas the whole-body LUBEL knockdown flies were more severely affected (see Fig 6C). We find this is an interesting point; for early time points,

LUBEL especially in muscle may have a major effect, but at later time points, LUBEL in other tissues is involved in the regulation of heat-shock dependent survival. We made this point clear in the text and rewrote the text.

The authors show a slight reduction of Hsp70 expression upon heat stress response in flies deprived of functional LUBEL for a single time point. Is this true over time, or is there only a slight delay in HSP70 gene expression?

We thank the reviewer by raising this interesting point. As the reviewer pointed out, it is also of our great interest to follow the dynamics of Hsp70 gene induction over time, especially at later time points. However, because of the early death of LUBEL mutant flies, starting around 60 min (as in Fig 6B), we could only test the earlier time point (30 min) to obtain RNAs in a reliable quality. We found that also at 30 min time point, the heat shock-dependent Hsp70 induction was significantly reduced in LUBEL mutant flies (CC/SS and delR2) (**a new Fig EV5G**). These data collectively indicate that at least at 30 min and 60 min time points, heat shock-dependent Hsp70 gene induction was significantly reduced in the LUBEL mutant flies.

Additionally, the RNaseq data presented in fig 6C are not really comprehensible and need reformatting for a better understanding of the figure.

We agree with the reviewer that the RNaseq data presentation in the original Figure 6C as well as in Figure 6D was not comprehensible. Statistical analysis based on ANOVA revealed that only 36 and 11 immune responsive genes out of 402 were identified as differentially expressed between any of the control and pricked sample groups, respectively. The majority of the immune responsive genes was expressed similarly between the sample groups, which indicates that there are no overall significant differences in immune responsive gene induction between control and LUBEL mutant flies. To better reflect this observation, we have changed the presentation style; we removed the heat maps, but instead added a list of those genes, which were identified to be significantly different (**new Tables EV1 and 2**).

2nd Editorial Decision

24 August 2016

Thank you for the submission of your revised manuscript to our editorial offices. I apologize getting back to you with delay, but due to the holiday season it took more time than expected to receive the referee reports, which you will find enclosed below. As you will see, referees #2 and #4 find the manuscript suitable for publication in EMBO reports. Referee #3 still doubts the physiological relevance of the described role of CG11321 during heat stress. Nevertheless, taking in consideration the positive assessments of referee #2 and #4, we think that your manuscript is now suitable for publication. However, referee #2 has two further concerns that we ask you to address during the final revision of your manuscript. We also ask you to consider the comments of referee #4 (and check if in the manuscript the "important role in the regulation of heat-tolerance" is not overemphasized - also not in the synopsis image) and his minor point.

Further, please update the callout of the appendix figures to just Figure S1, S2, etc. and Table S1 (not Appendix Figure S1) and please name the Table in the Appendix Table S1.

We also strongly encourage the publication of original source data with the aim of making primary data more accessible and transparent to the reader. The source data will be published in a separate source data file online along with the accepted manuscript and will be linked to the relevant figure. If you would like to use this opportunity, please submit the source data (for example scans of entire gels or blots, data points of graphs in an excel sheet, additional images, etc.) of your key experiments together with the revised manuscript. Please include size markers for scans of entire gels, label the scans with figure and panel number, and send one PDF file per figure or per figure panel.

I look forward to seeing a revised version of your manuscript when it is ready. Please let me know if you have questions or comments regarding the revision.

REFeree REPORTS

Referee #2:

In this revised version the authors have addressed the points of the reviewers in a satisfactory manner. Importantly, they could show that the background chain formation activity with the HECT cys mutant is normal for HOIP proteins, they reproduced the phenotype from a second muscle-specific allele, importantly, they showed that the effects on linear chains were not due to changes in expression of poly-ubiquitin besides a number of other improvements. With this it is a convincing analysis that shows an alternative function for *Drosophila* HOIP that may be important for the human allele as well (although so far buried in the immune response data) and will have broad interest.

There are two points in these new data that can be improved/adjusted

- a) the ANOVA analysis of the RNAseq is not explained and it is therefore unclear to me what is compared to what in the two respective tables. Moreover in the text the 36 and 11 genes between control and WT are mixed up, referring to 36 differences for control, whereas this is for the pricked. By selectively showing the immune responsive genes, the reader gets little insight in the importance of this difference, and of course the 25 extra genes are all in the bacterial response and does not now explain the text : "the analysis revealed that "immune response" scored low in differentially expressed biological processes.". It would be advisable to a) extend the legends to these tables and b) add another figure that shows why that statement was made
- b) it would be helpful to explain that the RING may be the origin of the background activity that is observed in the case of the HECT mutant. This would be different from the statement in the rebuttal that suggests it is E3-independent activity.

Referee #3:

The authors adequately addressed my concerns. Now, the reviewer believes that CG11321 is indeed ubiquitin ligase that generates linear ubiquitin chains. However, physiological importance and the molecular roles CG11321 played in heat shock responses have not shown.

Referee #4:

This version of the manuscript entitled "Linear ubiquitination by Lubel plays a critical role in *Drosophila* heat stress response" by Asoka et al., has now largely improved. The authors answered our questions and we particularly appreciate the clarification of the Lubel/linear ubiquitination role during the innate immune response. It seems now clear that Lubel has no function there.

Regarding the key finding of this study, the function of Lubel during the heat stress response in muscle, evidences are now convincing. However, the function Lubel seems restricted to a narrow period of time upon Heat shock and may explain why the authors struggled to identify a phenotype in Lubel deficient flies.

Collectively, this study represent an impressive piece of work and the conclusions seem now supported by data. The paper is very well written. However we still have some doubt concerning the global impact of this study, as mentioned in our initial report.

Minor point :
page 12 please change "diptericine" for "diptericin"

2nd Revision - authors' response

30 August 2016

We are very happy with referees' positive comments about our revised manuscript. We corrected points raised by the reviewers as below.

In short, in this revised manuscript,

- 1) We extended the legends for EV Table 1 and 2, and added new Tables (EV Table 3-8) for the RNA-Seq analysis to support our statement. We also modified and corrected the statement in the result section.
- 2) We confirmed that the LUBEL C2704A mutant is described to be the HECT mutant in which ubiquitin loading site is mutated. We also have mentioned that the residual activity for ubiquitination may derive from the RING activity.
- 3) We changed the text, the title and the synopsis figure not to over-emphasize about the role of LUBEL in the heat tolerance. Especially, we deleted the words, such as 'critical' and used the term 'heat response' instead of 'heat tolerance'.
- 4) We corrected minor mistakes in the text.

More detailed a point-by-point response can be found in the next pages.

We are confident that the manuscript is now strong and clear to be accepted to be published in the EMBO Reports.

We look forward to hearing a final decision.

We thank the referees for their positive responses. Please find below as our point-by-point response.

Referee #2:

In this revised version the authors have addressed the points of the reviewers in a satisfactory manner. Importantly, they could show that the background chain formation activity with the HECT cys mutant is normal for HOIP proteins, they reproduced the phenotype from a second muscle-specific allele, importantly, they showed that the effects on linear chains were not due to changes in expression of poly-ubiquitin besides a number of other improvements. With this it is a convincing analysis that shows an alternative function for *Drosophila* HOIP that may be important for the human allele as well (although so far buried in the immune response data) and will have broad interest.

There are two points in these new data that can be improved/adjusted

a) the ANOVA analysis of the RNAseq is not explained and it is therefore unclear to me what is compared to what in the two respective tables. Moreover in the text the 36 and 11 genes between control and WT are mixed up, referring to 36 differences for control, whereas this is for the pricked. By selectively showing the immune responsive genes, the reader gets little insight in the importance of this difference, and of course the 25 extra genes are all in the bacterial response and does not now explain the text : "the analysis revealed that "immune response" scored low in differentially expressed biological processes.". It would be advisable to a) extend the legends to these tables and b) add another figure that shows why that statement was made

We made more clear description about the RNA-Seq data analysis in the result section and corrected the mistake of the numbers 36 and 11 (page 12). As suggested a) we added extended legends for 2 tables (EV Table 1 and 2), and b) added new tables (EV Table 3-8) to explain about the immune response genes analyzed by the RNA-Seq.

More precisely, to clarify the statement of 'the analysis revealed that "immune response" scored low in differentially expressed biological processes', we modified the sentences and added tables (EV Table 3-8), which list the statistically and significantly enriched GO terms in all comparisons of control mutant vs. WT, and pricked mutant vs. WT. The pricked mutant vs. WT comparisons hadn't been performed earlier, so we have now performed these comparisons, filtered the DE genes with FC 2 and FDR 0.05, and then performed the enrichment analysis. We believe that with these data, our statement is supported well.

b) it would be helpful to explain that the RING may be the origin of the background activity that is observed in the case of the HECT mutant. This would be different from the statement in the rebuttal that suggests it is E3-independent activity.

We confirmed that we clearly state in the result section (page 7, in red) that the LUBEL C2704A mutant used for the in vitro ubiquitination assay in Fig 2A is a mutant in which ubiquitin-loading site is mutated. We also mentioned about the possible background activity, which may derive from the RING domains (page 7, in red).

Referee #3:

The authors adequately addressed my concerns. Now, the reviewer believes that CG11321 is indeed ubiquitin ligase that generates linear ubiquitin chains. However, physiological importance and the molecular roles CG11321 played in heat shock responses have not shown.

Referee #4:

This version of the manuscript entitled "Linear ubiquitination by Lubel plays a critical role in Drosophila heat stress response" by Asoka et al., has now largely improved. The authors answered our questions and we particularly appreciate the clarification of the Lubel/linear ubiquitination role during the innate immune response. It seems now clear that Lubel has no function there.

Regarding the key finding of this study, the function of Lubel during the heat stress response in muscle, evidences are now convincing. However, the function Lubel seems restricted to a narrow period of time upon Heat shock and may explain why the authors struggled to identify a phenotype in Lubel deficient flies.

We agree that LUBEL plays a role in the regulation of heat tolerance in rather a narrow period of time. We toned down about the statement of LUBEL-function in the heat tolerance in the main text as well as in the title.

Collectively, this study represent an impressive piece of work and the conclusions seem now supported by data. The paper is very well written.

However we still have some doubt concerning the global impact of this study, as mentioned in our initial report.

Minor point :
page 12 please change "diptericine" for "diptericin"

We thank the referee for picking this up. We corrected the typo in page 12, as well as in the figure EV4 G and its legend on page 46.

3rd Editorial Decision

05 September 2016

I am very pleased to accept your manuscript for publication in the next available issue of EMBO reports. Thank you for your contribution to our journal.

Corresponding Author Name: Fumiyo Ikeda

Journal Submitted to: EMBO Report

Manuscript Number: EMBOR-2016-42378-T